# Cubic 3D Chern photonic insulators with orientable large Chern vectors

Chiara Devescovi [1]✉, Mikel García-Díez[1,2], Iñigo Robredo[1,2], María Blanco de Paz[1], Jon Lasa-Alonso [1,3], Barry Bradlyn [4], Juan L. Mañes[2], Maia G. Vergniory[1,5,6]✉ & Aitzol García-Etxarri [1,5]✉

Time Reversal Symmetry (TRS) broken topological phases provide gapless surface states protected by topology, regardless of additional internal symmetries, spin or valley degrees of freedom. Despite the numerous demonstrations of 2D topological phases, few examples of 3D topological systems with TRS breaking exist. In this article, we devise a general strategy to design 3D Chern insulating (3D CI) cubic photonic crystals in a weakly TRS broken environment with orientable and arbitrarily large Chern vectors. The designs display topologically protected chiral and unidirectional surface states with disjoint equifrequency loops. The resulting crystals present the following characteristics: First, by increasing the Chern number, multiple surface states channels can be supported. Second, the Chern vector can be oriented along any direction simply changing the magnetization axis, opening up larger 3D CI/3D CI interfacing possibilities as compared to 2D. Third, by lowering the TRS breaking requirements, the system is ideal for realistic photonic applications where the magnetic response is weak.

[1] Donostia International Physics Center, Paseo Manuel de Lardizabal 4, 20018 Donostia-San Sebastián, Spain. [2] Physics Department, University of the Basque Country (UPV-EHU), Bilbao, Spain. [3] Centro de Física de Materiales, Paseo Manuel de Lardizabal 5, 20018 Donostia-San Sebastian, Spain. [4] Department of Physics and Institute for Condensed Matter Theory, University of Illinois at Urbana-Champaign, Urbana, IL 61801-3080, USA. [5] IKERBASQUE, Basque Foundation for Science, Maria Diaz de Haro 3, 48013 Bilbao, Spain. [6] Max Planck Institute for Chemical Physics of Solids, Dresden D-01187, Germany. ✉email: chiara.devescovi@dipc.org; maiagvergniory@dipc.org; aitzolgarcia@dipc.org

nspired by the discoveries of topological phenomena in solid state systems, the study of topology in the propagation of light in photonic crystals has been the subject of much recent attention[1–13]. Among all topological states of matter, time-reversal symmetry (TRS) broken topological materials, such as Chern insulators (CI)[2,3] and lasers[14], have been a particular focus due to their topologically protected unidirectional edge states with non-reciprocal propagation properties. In these systems, scattering processes from one boundary state into another are strongly suppressed, due to decoupling of counter-propagating 1D chiral edge channels[15,16].

Seminal works in 2D photonics have demonstrated that transverse magnetic (TM) modes in gyro-magnetic photonic crystals could mimic the Chern insulating state for light[4,5]. Due to a non-zero value of the topologically invariant Chern number, these systems were shown to sustain topologically protected one-way edge states with negligible dissipation and absence of back-scattering, even in presence of impurities and lattice defects which break translational symmetry.

As originally pointed out by Ref. [17], extending these ideas to 3D is in principle possible, under some more stringent conditions. As an example, preserving the translational symmetry of the lattice, a Chern insulating phase is predicted in 3D to host chiral anomalous surface states (SS) on its boundary[18–23]. In contrast to 2D, a 3D Chern insulator (3D CI) is a topological phase that can be characterized by three first Chern invariants–or a Chern vector $\mathbf{C} = (C_x, C_y, C_z)$ - defined on lower dimensional surfaces[24–26]: such a state of matter can support chiral surface states propagating on the planes with Miller indices indicated by the Chern vector.

Previous theoretical efforts in photonics[27] have engineered a TRS broken insulator with a single nonzero Chern number of unit value in a 3D uniaxial structure and employing a large magnetic field. However, the Chern vector was fixed to have a single component along a preferred axis selected by the fabrication, a situation similar to that of a stack of 2D Chern layers. Moreover, this design required strong TRS breaking, which is usually an arduous challenge in photonic crystals. Finally, the Chern number value was limited to unity.

At the same time, large Chern numbers have been observed in the band-gaps of 2D square photonic crystals[28,29]. Topological photonic systems with large Chern numbers can sustain multiple spatially separated edge states[29]. These edge states allow a plethora of applications, including unidirectional multimode waveguides where information can be multiplexed through the different edge states allowing for photonic on-chip communications with higher channel capacity[29]. Nevertheless, the value of the Chern number in these 2D systems was not determined by design, but a consequence of the particular system under study. In this sense, finding an engineering strategy to create photonic crystals with any given Chern number would be highly desirable and it is still a challenge that remains open.

In this work, we propose a method to design cubic 3D topological photonic crystals where Chern vectors of any magnitude, sign or direction can be implemented at will. Our method is based on the merging and annihilation of Weyl points through multi-fold supercell modulations in three dimensions[30]. As a result, we obtain a 3D CI phase with the following characteristics: First, since the Chern number is additive with respect to band-folding over large supercells, the system can support arbitrarily large Chern numbers and thus the coexistence of multi-channel unidirectional surface states. Second, owing to the cubic symmetry of the underlying modulated structure, the system can exhibit any combination of nonzero elements of a Chern vector, allowing for more 3D

CI/CI interfacing combinations as compared to 2D. Third, through the combined use of multi-fold supercells and reduced manipulation of Weyl points, the 3D CI phase can be realized under weak magnetization conditions, suited for realistic photonic applications. As a final step, we verify the emergence of chiral SS at interfaces between regions with differing Chern vectors, confirming the existence and spatial separation of unidirectional chiral partners. The outline of the paper is as follows: in the Results section, we provide full topological characterization of the bulk and the boundary of our photonic 3D CI. For the bulk, we show that the cubic system can support any nonzero element of the first Chern vector. The direction of the Chern vector can be tuned by changing the orientation of the applied external static magnetic field. We also prove a strategy for obtaining the 3D CI under a minimal magnetization and a method to design 3D CIs with specifically desired large Chern numbers. For the boundary, we demonstrate the emergence of unidirectional gapless SS and we analyze their 3D anomalous chiral properties. In the Methods section, we describe the numerical implementation of the technique by which we characterize the gap topology based on a 3D generalization of the photonic Wilson loop approach and provide a group theoretical analysis of the mechanism through which we create and manipulate the Weyl points and open up a topological gap by the use of supercell modulations. Further details on our design are given in the Supplementary Information.

## Results

The starting point of our design is a photonic crystal with a unit cell containing four dielectric rods directed along the main diagonals of a cubic crystal (scalable lattice parameter $|a|$). The rods meet at the origin of the unit cell, and the structure is invariant under the operations of the centrosymmetric and non-symmorphic space group (SG) $P\bar{n}3m$ (No. 224)[31]. Since we will later consider modulations of this structure, it is convenient to simulate the dielectric rods by assembling dielectric spheres with radius $r = r_0$ along $(x, y, z)_0/|a| = (t, t, t), (t, 1-t, 1-t)$, $(1-t, t, 1-t), (1-t, 1-t, t)$ with $0 < t < 1/2$, i.e. employing a spherical covering approximation[32]. The resulting design is shown in Fig. 1a. To obtain a TRS-preserving system, the dielectric material is described by a diagonal (isotropic) permittivity tensor, $\varepsilon_{TRS} = \varepsilon \mathbb{1}_3$, where $\mathbb{1}_3 = \hat{\mathbf{x}} \otimes \hat{\mathbf{x}} + \hat{\mathbf{y}} \otimes \hat{\mathbf{y}} + \hat{\mathbf{z}} \otimes \hat{\mathbf{z}}$ and $\varepsilon \in \mathbb{R}$ (no losses), and by unit magnetic permeability $\mu = \mathbb{1}_3$. To simulate the optical response of the system, we employ the MIT Photonic Bands (MPB) software package[33]. As shown in Fig. 1a, with TRS, the photonic band-structure presents a three-fold degeneracy between the three lowest energy bands at the high symmetry point $\mathbf{R} = \frac{2\pi}{|a|}(1/2, 1/2, 1/2)$; note that, in the displayed energy window, the two lowest bands are fully degenerate. Everywhere else in the Brillouin zone (BZ), there is a gap between the second and third band. The dispersion reflects the three-fold rotational symmetry of the crystal, and so is invariant under cyclic permutations of the three $k_i$ ($i = x, y, z$) directions. In order to keep the notation consistent throughout the paper and to capture all the variety of symmetry designs, we label the high symmetry points in the BZ according to the convention described in the Supplementary Note (SN) 5, i.e. in a Cartesian orthorhombic convention.

Following a strategy introduced in[31,34,35] we break TRS to split the three-fold degeneracy at $\mathbf{R}$ into two Weyl points[31]. This can be achieved by either applying an external magnetic field bias to a gyro-electric crystal[2,3] as we proceed here, or by exploiting the internal remnant magnetization of ferri-magnetic materials[5]. TRS breaking is implemented by introducing off-diagonal imaginary elements in the permittivity tensor: for the specific case of an applied magnetic field $\mathbf{B} = B_z\hat{\mathbf{z}}$, the induced gyro-electric

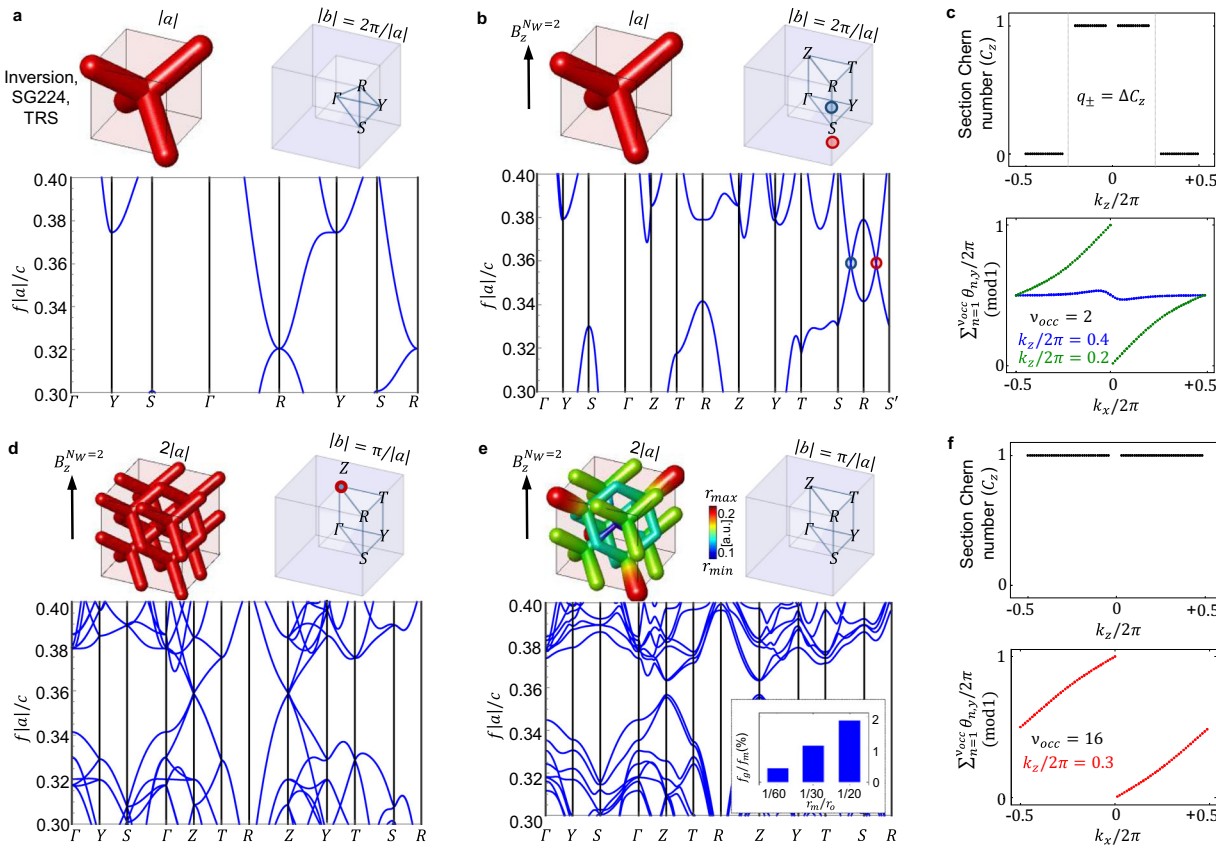

**Fig. 1 Photonic 3D CI by cubic supercell modulation at $N = N_W = 2$.** Each (**a**, **b**, **d**, **e**) panel shows the crystal unit cell, the irreducible Brillouin zone (IBZ) and the band-structure (BS). Frequencies $f$ are given in reduced units, $|a|$ being the scale invariant lattice parameter and $c$ the speed of light. Sectors (**c**) and (**f**) contain the topological characterization via photonic Wilson loops (WL), in the Weyl semimetallic (WS) phase and in the 3D CI phase, respectively. **a** Photonic crystal constructed from cylinders of radius $r_0 = 0.15$ and dielectric constant $\varepsilon = 16$ at TRS. The three lowest photonic modes display a three-fold degeneracy at **R** and the two lowest bands are fully degenerate in the displayed energy window. **b** TRS breaking implemented via a gyro-electric response with $\eta_z^{N_W=2} = 16$: the bias field is adjusted in order to split the Weyl points of approximately half the BZ, i.e. at $k_z^\pm = \pm\frac{\pi}{2|a|}$, along the **SRS'** line where $\mathbf{S'} = \mathbf{S} - \mathbf{b}_z$. **c** Electromagnetic section Chern number calculated on 2D $k_z$ planes normal to the magnetization (upper panel) and transverse flow of the $\theta_y$ eigenvalue of the WL matrix summed over the entire subset of $\nu_{occ}$ bands lying below the local gap at $k_z$ (lower panel). The discontinuity of the section Chern number at the wavevector of each Weyl point $k_z^\pm$ is used a measure its topological charge ($q_\pm = 1$). **d** Artificial folding of the bands on a $N = 2$ cubic supercell: Weyl points superimpose at **Z** in the new BZ. **e** Coupling and annihilation of Weyl points through a $N = 2$ supercell modulation with parameter $r_m = r_0/20$. The amplified modulation is graphically visualized via a colorbar with $r_{max} = r_0 + r_m$ and $r_{min} = r_0 - r_m$. A topological direct gap ($f_g$) at **Z** is opened, with gap-to-midgap ($f_g/f_m$) ratio of 1.86%. The size of gap can be appropriately tuned by choosing the value of the modulation, as in the inset. **f** The section Chern $C_z$ number displays constant unit value everywhere in the BZ, establishing the system to be in the 3D CI phase. Source data are provided as a Source Data file.

tensor is:

$$\varepsilon_{\eta_z} = \begin{pmatrix} \varepsilon_\perp & i\eta_z & 0 \\ -i\eta_z & \varepsilon_\perp & 0 \\ 0 & 0 & \varepsilon \end{pmatrix}, \qquad (1)$$

where $\eta_z = \eta_z(B_z)$ the bias-dependent gyro-electric parameter and $\varepsilon_\perp = \sqrt{\varepsilon^2 + \eta_z^2}$. The gyro-electric tensor corresponding to magnetic fields in other directions can be obtained by orthogonal rotations of Eq. (1).

Under these conditions, the three-fold degeneracy splits into a pair of Weyl points (or a Weyl dipole). In order to analyze the formation and splitting of the Weyl points under TRS breaking and to predict the direction of their displacement in the BZ, we develop a $\mathbf{k}\cdot\mathbf{p}$ model around the three-fold degeneracy at **R**. The model, based on the group theoretical method of invariants (see Methods), allows us to conclude that Weyl points appear at inversion-symmetric positions with respect to **R** along the $k_z$

direction, with a separation that can be adjusted by choosing the bias field $B_z$ appropriately. Our MPB simulations, presented in Fig. 1b, confirm this predictions accurately. More generally, for a magnetization applied along any of the main coordinate axes $x_i$, the Weyl dipole is oriented along the line joining **R** to $\mathbf{R'} = \mathbf{R} - \mathbf{b}_i$, where $\mathbf{b}_i$ is the corresponding primitive reciprocal lattice vector. For a detailed comparison of the analytic model and the numerical simulations see Methods.

Next, we calculate the chiral topological charge $q_\pm$ of the Weyl points in the $\mathbf{k}\cdot\mathbf{p}$ model using the Z2Pack numerical tool[36,37], concluding that the Weyl points have opposite valued unit charges ($q_\pm = \pm 1$).

We confirm these predictions by computing the topological charges directly from the MPB eigenstate solutions. To do so, we implement a numerical approach based on the analysis of the winding properties of photonic hybrid Wannier energy centers (WEC). WEC are accurately defined in the Methods section. There, we establish a mapping from electronic Wannier charge

centers (WCC)[38,39] to photonics and we perform a generalization of the photonic Wilson loop approach of Ref. [40] (initially implemented for 2D scalar waves) applicable to fully 3D electromagnetic (EM) vector fields. The results of this analysis are summarized in Fig. 1c: the top panel shows the electromagnetic Chern number $C_z$ of the two lowest bands calculated on 2D planes orthogonal to the magnetization axis. We observe a sharp discontinuity $\Delta C_z = \pm 1$ at the wavevector of each Weyl point. This is similarly reflected in the winding of the Wilson loop eigenvalues on two selected planes, as shown in the bottom panel. From the discontinuity in the section Chern number at each Weyl point, we deduce the associated topological charge[41], confirming that $q_\pm = \pm 1$, as predicted by the $\mathbf{k} \cdot \mathbf{p}$ model.

In order to identify a geometrical perturbation able to open a band-gap, we analyze the coupling of the Weyl points from a group theoretical perspective, performing a generalization of the method of invariants now suited to capture translation breaking perturbations (see Methods). We conclude that the only deformation of the geometrical structure leading to their annihilation and to the opening of a topological gap are lattice commensurate supercell modulations. In particular, we find that it is possible to independently activate the supercell modulation along the $x_i$ Cartesian directions and couple Weyl points generated by the corresponding magnetic field $B_i$. Note that, from our WEC analysis, we see that each BZ plane between the Weyl nodes carries Chern number $|C_z| = 1$, while each plane outside the Weyl nodes carries Chern number $|C_z| = 0$. When we couple the Weyl nodes by a lattice commensurate modulation, we backfold the BZ into a region commensurate with the Weyl node separation vector, which is a reciprocal lattice vector in the folded BZ. The additivity of the Chern number then ensures that every plane in the reduced BZ carries a nonzero Chern number, resulting in a 3D CI when the gap is opened[30]. This expresses the fact that the Chern number density of our 3D system does not change as a function of the (TR-even) supercell modulation; it simply goes from being unquantized in the original system (necessitating the existence of Weyl points), to being a quantized multiple of a reciprocal lattice vector in the modulated system.

With this starting setup, in order to obtain 3D Chern insulating phases, we will follow a general three-step strategy:

1.  First, using the external magnetic field we move the Weyl points at fractional distances of the Brillouin zone (BZ), i.e. at positions $\mathbf{K}_{1,2} = \mathbf{R} \pm \frac{\mathbf{X}_i}{N_W}$ where $N_W \in \mathbb{N}$ and $N_W > 1$. In this way, in a further step, we will be able to couple and gap the Weyl points with a commensurate modulation of a supercell structure. Notice that larger $N_W$ are associated to smaller splittings.

2.  Secondly, we fold the BZ by creating multi-fold $(N > 1)$ supercells; this is achieved by replicating the original unit cell either in a cubic supercell of dimensions $(N, N, N)$ or in a uniaxial supercell of size $N$ directed along the magnetic field direction. This step of the procedure will merge the Weyl points, originally at $\mathbf{K}_{1,2}$ in the natural BZ, to the same $\mathbf{k}$ point in the new reduced BZ, forming a four-fold degeneracy. In the SN 2, we show that fine tuning and perfect band folding are not strictly necessary for opening a gap at the Weyl points. This endows our system with a robustness and tolerance against reciprocal lattice vector mismatches.

3.  As a third and last step, we couple and gap the opposite-charge Weyl points by spatially modulating the crystal geometry with a periodicity commensurate to the designed supercell. More specifically, we vary the radius of the cylinders through the entire supercell: numerically, this is achieved by locally changing the radius of the spheres in the

covering approximation, from their original $r_0$ radius to the new local one $r(x, y, z)$. Coherently to the choice made in the previous point 2, this is either done with a cubic modulation of the type: $\Delta r(x, y, z) = r(x, y, z) - r_0 = r_m[\cos(2\pi x/N|a|) + \cos(2\pi y/N|a|) + \cos(2\pi z/N|a|)]$ when all the Cartesian components of the modulation are turned on or with a uniaxial modulation, where only the component oriented along the magnetic field is activated, e.g. $\Delta r(x, y, z) = r_m \cos(2\pi x_i/N|a|)$ for a field with $B_i \neq 0$ field. More details are given in the SN 6.

Depending on the values of the parameters $N_W$ and $N$, it is possible to design different tailored 3D CI phases, in particular: a cubic 3D CI with orientable Chern vectors, a 3D CI in a reduced magnetization environment and a 3D CI with tunable larger Chern numbers.

We stress that the argument of gap opening by folding and supercell modulation is very general, and can be applied as long as the constraints of commensurability between the Weyl diplacement and the supercell size are satisfied. Therefore, any other crystal structure exhibiting a pair of Weyl points could be perfectly suited to their annihilation via the mechanism proposed.

**Cubic 3D CI.** Our first objective is to design a cubic 3D CI with orientable Chern vectors. As we will show, this can be achieved using a cubic supercell modulation with $N = N_W > 1$. In order to keep the MPB simulations computationally affordable we consider the simplest case of $N = N_W = 2$, which requires to separate the Weyl points to half the BZ as in Fig. 1b. The effect of band folding in such a system is visualized in Fig. 1d: for a field oriented as $B_z$, the two Weyl points superimpose to form an artificial four-fold degeneracy at $\mathbf{X}_3 \equiv \mathbf{Z}$. More generally, on a $N = N_W$ cubic supercell and from simple folding considerations, we expect the opposite-charge Weyl points to merge at $\mathbf{X}_i$ when $N = N_W$ is even, at $\mathbf{R}_i - \mathbf{X}_i$ when $N = N_W$ is odd, where $\mathbf{X}_i = \frac{\mathbf{b}_i}{2} \equiv \mathbf{X}, \mathbf{Y}, \mathbf{Z}$. From this starting point, in order to realize a cubic 3D CI phase with orientable Chern vectors, all the three Cartesian components of the cubic commensurate modulation need to be simultaneously turned on. The resulting photonic band structure of the $N_W = N = 2$ supercell modulated structure is shown in Fig. 1e: as it can be seen, the Weyl points annihilate and open up a gap. To numerically verify the topological properties of this bulk gap in our design we compute photonic Wilson loops and analyze their winding in the BZ (see Methods). Our results, summarized in Fig. 1f, determine that the obtained insulating phase acquires a nonzero Chern number along every plane perpendicular to the magnetization axis as predicted by the $\mathbf{k} \cdot \mathbf{p}$ model. Therefore, by simply changing the orientation of the magnetization axis, it is possible to select each Cartesian component in a first Chern class vector $(C_x, C_y, C_z)$, due to the the cubic nature of the underlying system and modulation. Note that the existence of three weak indices in 3D allows for more interfacing possibilities as compared to the 2D case, where only the trivial/TI and the opposite (or different) Chern number combinations are realizable, as discussed later.

**3D CI at reduced magnetization.** In our previous example we required the Weyl points to be displaced to the half of the BZ. Achieving such a condition requires large TRS breaking parameters. In our simulations, fulfilling this requirement implied using a magnetization bias corresponding to $\eta^{N_W=2} = 16$. Note that, to date, large gyrotropic parameters have been experimentally achieved in photonic crystals only in the microwave frequency regime via ferri-magnetic materials[5,42] and that the gyrotropic response of most currently known dielectric materials

| **Table 1 Reduction of the magnetic bias $\eta$ and of the topological gap-to-midgap ratio $f_g/f_m(\%)$, still achieving the same Chern number $C$. Source data are provided as a Source Data file.** | | |
|---|---|---|
| $N^a = N_W^b$ | 2 | 3 |
| $C^c$ | 1 | 1 |
| $\eta^d$ | 16 | 7.8 |
| $f_g/f_m(\%)^e$ | 1.5 | 1.2 |

<sup>a</sup>Supercell size.
<sup>b</sup>Weyl Dipole splitting fraction.
<sup>c</sup>Chern number
<sup>d</sup>Gyrotropic parameter.
<sup>e</sup>Topological gap-to-midgap ratio.

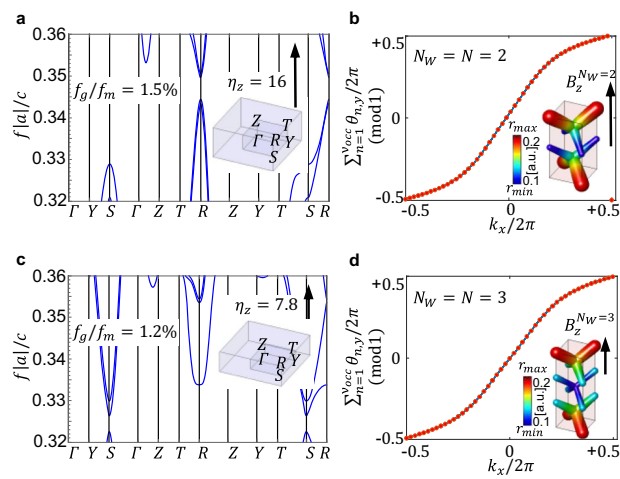

**Fig. 2 3D CI in a reduced magnetization environment.** Uniaxial supercells $(1, 1, N)$ with modulation parameter $r_m = r_0/20$ and WLs on selected planes for $k_z/2\pi = 0.3$. **a**, **b** 3D CI with $C_z = 1$ at large magnetization $\eta^{N_W=2} = 16$, corresponding to the maximum Weyl dipole separation and band-gap. **c**, **d** 3D CI in a reduced magnetic environment $\eta^{N_W=3} = 7.8$. The band-gap only suffers a moderate contraction, yet the Chern vector and the topological properties are preserved. Source data are provided as a Source Data file.

is weak. Therefore, in this section, we suggest a way to hugely reduce the magnetization requirements for obtaining CI phase by employing multi-fold supercells and by increasing the intensity of the modulation. Instead of displacing the Weyl points to half the BZ and applying a supercell modulation over two original unit cells, we now move the Weyl points to a smaller fractional distance of the BZ and apply a supercell modulation over a larger number of original supercells to merge and gap the Weyl points appropriately. The resulting 3D CI phase still displays the same Chern number as in the maximally TRS broken system with $N = N_W = 2$, but it occurs in a largely reduced magnetic field environment due to the smaller **k**-space displacement of the Weyl points.

For example, making $N = N_W = 3$, which corresponds to a dipole separation of one third of the BZ and spatially modulating the structure over 3 original unit cells, one can get the same topological phase as in the $N = N_W = 2$ example. In order to keep the calculations computationally feasible, we simulate a uniaxial system of size $(1, 1, N)$. Nevertheless, the concept is readily generalizable to cubic supercells. Under this construction, the CI phase is achieved at $\eta_z^{N_W=3} = 7.8$. As it can be naturally expected, the resulting 3D CI suffers a moderate reduction in the band-gap. However, as shown in Table 1 and Fig. 2 where we compare the designs at $N = N_W = 2$ and $N = N_W = 3$, the later design presents a good compromise between gap size and TRS-breaking amplitude, considering the large advantage coming from a large drop in the required magnetization bias (from $\eta^{N_W=2} = 16$ down to $\eta^{N_W=3} = 7.8$). This could be of particular interest for photonic applications where the magnetic response is weak. The gyrotropic parameter value can be further decreased, as shown in the SN 3, by modulating over even larger supercells $N_W = 5, 6, 7$ and by optimizing the modulation intensity in order to partially compensate for the band-gap decrease. However we cannot indefinitely iterate this procedure down to zero bias since a compromise on the band-gap is always unavoidable: indeed, in the limit of very large $N$, there is no splitting of Weyl points and thus no TRS broken gap can clearly be opened, $\lim_{N\to\infty} f_g = 0$.

Figure 2 and Table 1 show the drop in the magnetization as compared to the decrease in the topological gap computed for uniaxial supercells at $N = 2$ and $N = 3$ with the same supercell modulation parameter $r_m$.

**3D CI with larger Chern numbers**. Lastly, we show that our design strategy can also be used to design photonic TIs wih larger Chern numbers. This can be achieved by modulating over even multi-fold supercells with $N = 2n > 2$, $n \in \mathbb{N}$, while keeping $N_W = 2$. The use of larger supercells permits folding of the BZ multiple times. In the band-folding process the Chern number contribution in each folded region of the BZ adds up. We thus

expect the gap resulting for such a modulated system to achieve larger Chern numbers $C_i$ according to the following relation: $C_i = n$. To prove this, we build uniaxial supercells of size $(1, 1, 2n)$ with $n = 1, 2, 3, 4$. These crystalline supercells are magnetized along the $\hat{\mathbf{z}}$ direction, creating Weyl points at half of the original BZ ($N_W = 2$). After folding, we find that the Weyl points are superimposed at $\mathbf{R} - \mathbf{Z} \equiv \mathbf{S}$ if $n$ is even and at $\mathbf{R}$ if $n$ is odd. As a final step we activate the modulation along the $z$ direction. In Fig. 3 we then calculate the Chern number of the band gaps in these systems using photonic Wilson loops (WL) by analyzing their winding in the BZ. The modulated supercells with $n = 1, 2, 3, 4$ under the appropriate TRS breaking acquire, as predicted, Chern numbers $C_z = 1, 2, 3, 4$ respectively. We restrict our calculations of uniaxial systems due to computational limitations, nevertheless, the argument for the Chern number growth holds similarly in the cubic case, when all the three components of the modulation are turned on.

**Chiral surface states**. Finally, to characterize the bulk-boundary correspondence in the designed systems, we now analyze the emergence of SS at the interface between the cubic 3D CI and a trivially gapped photonic crystal. Other CI/CI interfacing possibilities are discussed in the SN 5. Finding a proper insulating interface is an important requirement to prevent propagation of edge modes in free space due to modes living in the light cone. Furthermore, we also must avoid the formation of dangling defect states due to lattice mismatches. This is usually a quite difficult task in 3D, due to limited available band-gap geometries as compared 2D (details on the trivial interface in the SN 8). To keep the simulations numerically affordable, we stick to the case of a cubic supercell with $N = N_W = 2$ and analyze a topological slab with normal vector oriented along $\hat{\mathbf{x}}$, in presence of a $B_z$ field. From the bulk-boundary considerations, we expect unidirectional chiral SS to appear on the planes parallel to the magnetization (i.e., with normal vectors perpendidcular to the magnetization direction). Surface states are considered unidirectional in the following sense: The component of the group velocity (or

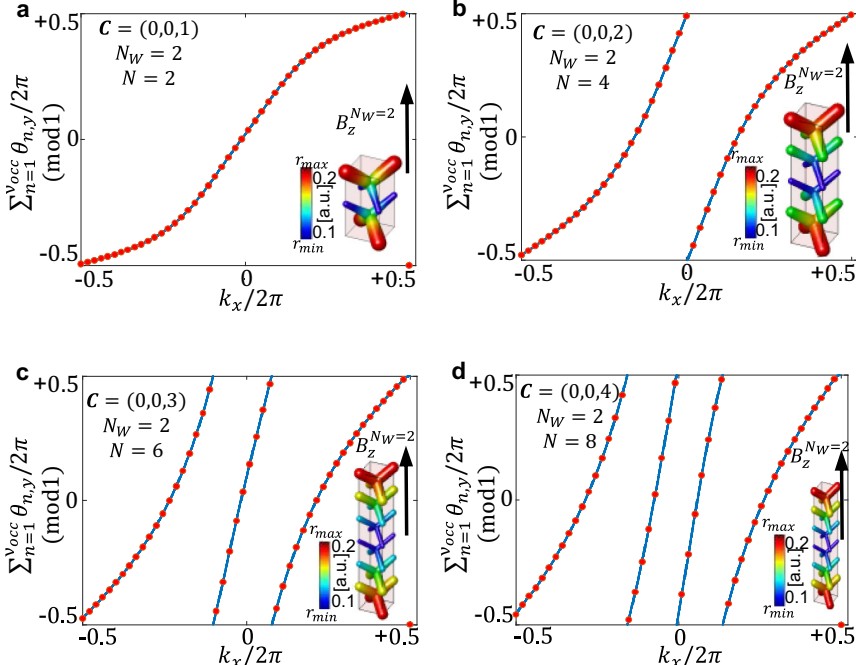

**Fig. 3 Generating larger Chern numbers.** Annihilation of Weyl points at $N_W = 2$ over multi-fold supercells of even $N$. **a** Unit Chern number 3D CI. **b**, **c**, **d** Increasing Chern numbers $C$ according to the relation $C = N/2$. Calculations performed on 1D supercells $(1, 1, N)$ with modulation parameter $r_m = r_0/20$ and WL on selected planes for $k_z/2\pi = 0.3$. Source data are provided as a Source Data file.

Poynting vector) normal to the magnetization direction has a well defined sign i.e. surface states cannot back-scatter along this specific direction. This component will be later denoted as conserved component.

With this setup, we characterize the hallmarks of chiral SS propagation using a combined real-reciprocal space analysis. Fig. 4a shows the band-structure for the (100)-surface, confirming the emergence of chiral SS connecting the lower and upper bands and fully crossing the band-gap. To better visualize the SS energy dispersion, in Fig. 4b we consider a 3D surface plot, out of which we take the midgap equifrequency cut shown in Fig. 4c. We observe the emergence of 3D Chern Fermi loops which are the natural evolution of the Fermi arcs of the photonic Weyl semimetallic phase. In the SN 11, we show how the SS of the WS phase evolve into the 3D Chern SS as an consequence of Weyl points annihilation. As we will show now, these Fermi loops can be separated in real space, i.e. we can associate those with positive group velocity component normal to the magnetization to a surface of the slab and those with negative one to the other surface. We will therefore consider them as disjoint. In order to establish the relation between counter-propagating modes with respect to the direction orthogonal to the magnetization axis $\hat{\mathbf{z}}$ and the interface normal $\hat{\mathbf{x}}$, i.e. $\hat{\mathbf{y}} = \hat{\mathbf{z}} \times \hat{\mathbf{x}}$, we analyze the propagation of individual edge channels at fixed $k_z$. As indicated by red/blue colors in Fig. 4(b, c), modes propagating with positive transverse group velocity $v_y > 0$ appear on one side of the topological slab, their flow being compensated by counter-propagating $v_y < 0$ partners located on the other surface of the slab. We define as chiral partners, the pair of surface states living on the opposite sides of the slab, moving with opposite component of the group velocity which is normal to magnetization axis. This feature is visualized in Fig. 4(d, f) where we select a pair of chiral partners for explanatory purposes and display their electric field profile in real space on cross sectional view of the crystal slab. We conclude that each disjoint piece of the SS energy sheet in Fig. 4c corresponds to $v_y > 0$ and $v_y < 0$: this spatial separation of chiral partners, provided by the bulk, is the

protection mechanism which prevents the back-scattering of one state into the other. Because of this, the presence of touching points in the SS dispersion between different chiral partners (red and blue lines in Fig. 4c), is purely accidental. As so, these crossings occur between states that reside on opposite sides of the slab and are physically separated in real space by the bulk. Therefore they cannot gap out, up to exponentially small finite size effects, and are protected by the spatial separation separation of chiral partners on opposite surfaces.

Furthermore, in order to investigate the possibility of energy propagation along the magnetization axis, we analyze the Poynting vector associated to our SS. We find that, even if individual edge channels display nonzero propagation along the magnetization direction, e.g. as in Fig. 4f, integrating the total contribution of entire SS yields no net energy transport along the bias field, as expected due to equilibrium conditions (see SN 9). Interestingly, analyzing the polarization state of each edge channel, we observe a well defined sign of the spatially-averaged optical chirality $\bar{c} > 0$[43–48] through the entire SS (see SN 10).

## Discussions

In this work, we developed a strategy to induce annihilation of Weyl points through cubic and multi-fold supercell modulations, allowing us to achieve a photonic 3D CI phase with the following characteristics:

First, arbitrarily large Chern numbers can be achieved by design, allowing for multi-modal propagation of topological surface states[28[,29]. On the one hand, the system with Chern number $N$ supports $N$ equifrequency loops. These $N$ equifrequency loops are compressed into a folded BZ that is $1/N$ the size of the original BZ. In this sense, if we are interested in quantities integrated over the BZ, we cannot expect an increase in extensive quantities such us the total field intensity. However, if we are interested in addressing states at a particular wavevector, which is a reasonable constraint in photonic systems, then the modulation has allowed us to address $N$ chiral surface modes with equivalent reciprocal

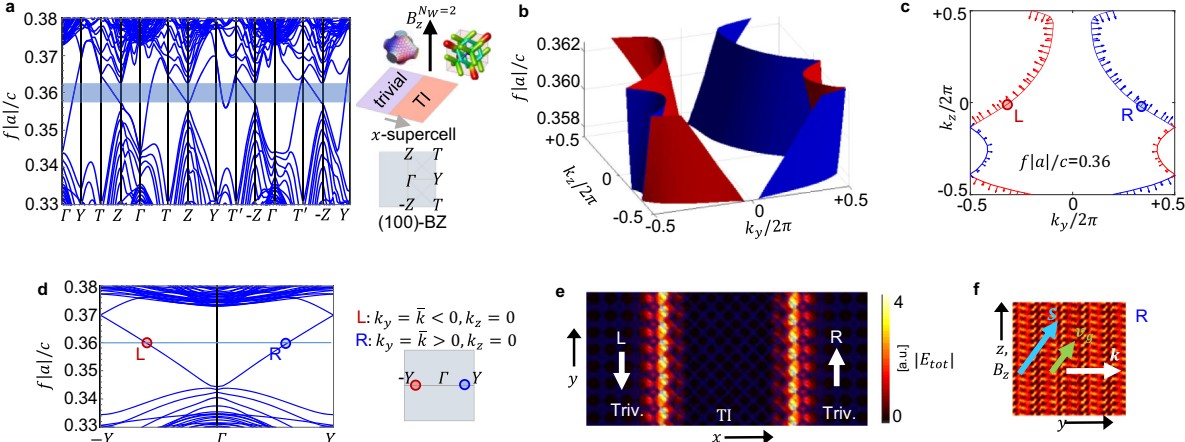

**Fig. 4 3D CI surface states.** Projected band-structure on the surface BZ with Miller index (100), for the interface between $N_{triv} = 8$ cells of trivial insulator and $N_{TI} = 8$ cells of TI, under a $\mathbf{B} = B_z \hat{\mathbf{z}}$ field and with unit Chern number. Edge states cross the topological gap, highlighted in the shaded region. The extended BZ is fully displayed from $-\mathbf{Z}$ to $\mathbf{Z}$, since it lacks of TRS due to application of a $\hat{\mathbf{z}}$ directed bias field. **b** 2D dispersion of topological surface states on the projected BZ, in the energy range of the topological band-gap. **c** Equifrequency loops for a cut taken at midgap: the arrows on the plot indicate the direction of the group velocity $\mathbf{v}_g = \nabla_{\mathbf{k}} f |a|/c = (v_y, v_z)$. Blue and red colors correspond to chiral partners with opposite $v_y$ and same optical chirality $\bar{c}$ which are located on opposite left/right sides of the interface $(L, R)$. **d** Edge states dispersion along a direction normal to both the interface and the external magnetic field; highlighted in circles is a pair of counterpropagating $m - th$ edge channels with $\pm k_y$: the spatial profile of their total electric field is shown in two following panels. **e** Counterpropagating chiral partners located on the two opposite left/right surfaces of the sample $(L, R)$: wavefront propagation in the $\mathbf{k}$ direction indicated by White arrows on a $xy$ cross section. **f** Spatial profile of the total electric field on a $yz$ planar cut: green and blue arrows indicate respectively the group velocity $\mathbf{v}_g$ and the total Poynting vector $\mathbf{S}$, the last being obtained from a spatial integral of the quantity $Re(\mathbf{E} \times \mathbf{H}^*)$, calculated directly from the EM field profiles. Differently from what expected from a conventional QHE edge channel, the Poynting vector displays a component parallel to the external magnetic field ($S_z \neq 0$), suggesting both QHE and CME features and the possibility of energy flow along the magnetization axis: we excluded such a possibility in the SN 9, by summing up the contributions of all the edge channels constituting the surface state. Source data are provided as a Source Data file.

lattice vectors, i.e. achieve unidirectional multiple surface mode operation. The capability of designing photonic systems with large Chern numbers in 3D could find interesting applications in the development of the emergent field of topological lasers[14,49] with a larger number of unidirectional SS.

Second, we showed that any element of the first Chern class vector can be selected by simply changing the magnetization direction, allowing for unique 3D CI/3D CI interfacing combinations as compared to 2D. For example, considering that every planar cut parallel to the magnetization direction is capable of supporting anomalous surface states, it could be worth investigating what occurs at a 3D $C_yI$/3D $C_zI$ planar interface. Similarly, owing to cubic symmetry of the 3D CI system, it would be possible to tune the relative angular phase in the supercell modulation in order to either break spatial inversion or not, which could lead to interesting surface states at the boundary of an obstructed atomic insulator (OAI) 3D CI and a 3D CI[30]. This may allow to develop of interesting photonic analogues of axionic responses[26,30] that we are investigating as part of a future work. Another possibility allowed just in 3D, could be to arrange different 3D CIs around an inert core, with the 3D CI composing each panel having Chern vector $(C_x, C_y, C_z)$ oriented to point inwards (e.g. fixing a 3D + $C_xI$ on a left $\hat{x}$ panel). Such a 3D interfacing arrangement, originally proposed in Ref. [26] as a possible realization of a magneto-electrical (ME) coupler in the field of electronics, has not yet a realization or equivalent in photonics. Analysis of all these challenging designs is left for further investigation.

Third, we showed the TRS breaking parameters required to induce this 3D CI phase can be substantially diminished by the use of larger supercells, which can enable the realization of a 3D CI phase also in photonic systems where the magnetic response is weak or it is not possible to manipulate largely the Weyl points in

the BZ. We also intend to emphasize that the strategy we devised in our paper is material agnostic, and can be easily adapted to any to-be-discovered experimental platform. In that sense, our work provides a roadmap to future experimental exploration of topological photonic crystals by showing how to reduce the needed magnetic response.

Finally, we showed that 3D CI photonic phase obtained displays chiral surface states on the planes orthogonal to the magnetization. As a remarkable signature of this, we observed the formation of disjoint equifrequency loops structures associated to the spatial separation of optically-chiral and counter-propagating partners. In conclusion, our system provides a realization of a photonic 3D CI state of matter in a fully cubic platform, with large Chern vectors engineered by design and in a weakly magnetic environment.

## Methods

**EM section Chern number from Wilson loop approach in 3D.** In order to provide a topological characterization of the bulk of the 3D CI photonic phase, we employ a mapping between electronic hybrid Wannier charge centers (WCC)[38,39] and photonic hybrid Wannier energy centers (WEC) in 3D. Hybrid WEC of the type $\theta_{n,y}$, which are localized in the $y$-direction and flowing in the $k_x$ transverse direction[38], are computed for each fixed $k_z$ plane from the subset of $n = 1, \ldots, N_b$ bands below the local gap at $k_z$, as follows[39]:

$$\theta_{n,y}(k_x) = -\text{Imlog}(w_n(k_x)), \quad (2)$$

where $w_n$ are the eigenvalues of the Wilson loop (WL) operator. The real space correspondence in terms of the $i$-th lattice parameter $|a_i|$ is given by $2\pi WEC_{n,i} \equiv |a_i|\theta_{n,i}$. As shown in[50,51], the WL operator can be numerically implemented as a path-ordered product of overlap matrices

$$\hat{W}(k_x) = \prod_{k_{y_i} \in l} \hat{M}^{k_{y_i}, k_{y_{i+1}}}, \quad (3)$$

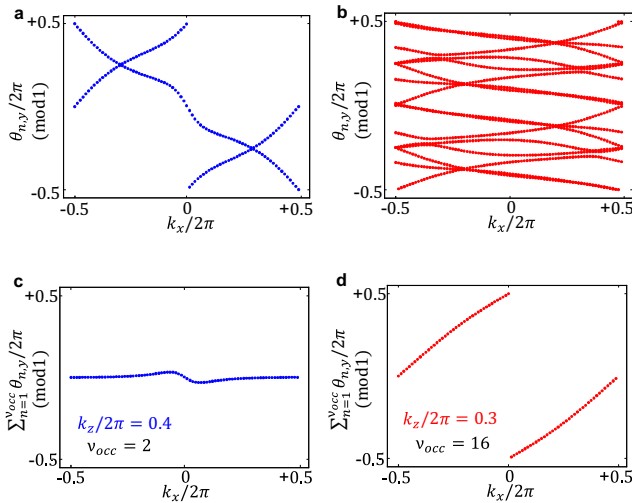

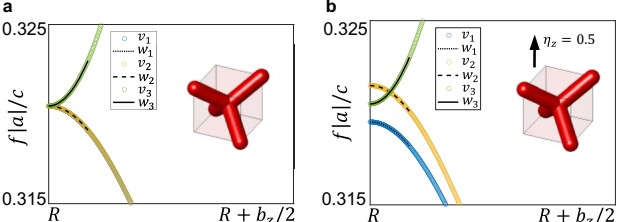

**Fig. 6 Comparison of the numerical band-dispersion with the analytical model.** Extracting the coefficients of the analytical $k \cdot p$ model in the vicinity of $\mathbf{R}$. Here, we are considering an extrapolation of the model at a distance of $\delta k = 0.008$ from $\mathbf{R}$. The empty circles are for numerical computed bands $v_{1,2,3}$ while the lines for the analytical dispersion $w_{1,2,3}$. **a** In presence of TRS, we conclude that $b_0 = -2.9a_0 > 0$. **b** In a weak field $\eta = 0.5$, we obtain that $\alpha_0 \sim -\beta_0$ since the third band does not move in energy and $\alpha_0 \sim 0$ since the vertical displacement of the two lowest degenerate modes is equal and opposite. Weyl points appear at positions: $k_z^\pm = \pm \sqrt{\frac{|\delta_0 B_z|}{b_0}}$, which here is at $k_z^\pm = \pm 0.004$. The $\delta_0 B_z > 0$ and $\delta_0 B_z < 0$ cases are equivalent upon reversing the two lowest photonic modes. Source data are provided as a Source Data file.

Both analytical models are based on the standard group theoretical method of invariants, which can be found in the literature[57]. This allows us to find an expansion in powers of the wave vector $\mathbf{k}$ of the photonic energy bands $\omega$, able to replicate the photonic modes dispersion. In order to do so, we construct an effective energy dispersion operator $H(\mathbf{k})$, expressed in terms of the space group irreducible representations bases, able to capture the photonic modes symmetry properties. In the photonic context, $H(\mathbf{k})$ can be viewed as a perturbative expansion of the Maxwell-Bloch operator acting on the electromagnetic fields in the first order formulation of Maxwell's equations[58]. For other applications of this approach to photonic systems, see e.g. Refs. [3,59].

In what follows, we adopt the notation convention taken in the Bilbao Crystallographic Server (BCS)[60], unless otherwise stated, and express reciprocal lattice vectors in reduced units $2\pi/|a| = 1$.

*Threefold degenerate model at R.* The following model describes the local behavior of the modes which are threefold degenerate at point $\mathbf{R} = (1/2, 1/2, 1/2)$ (see Fig. 6(a)). From the numerical computations, we know that this degeneracy is related to the three dimensional small representation $R_4^-$ of the little group of $\mathbf{R}$, given the transformation properties of these modes under the elements of the space group $Pn\bar{3}m$ (No. 224).

Following the method of invariants, we first note that the product $R_4^{-*} \times R_4^-$ is decomposed into small irreducible representations (irreps) at $\Gamma$ as:

$$R_4^{-*} \times R_4^- = \Gamma_1^+ + \Gamma_3^+ + \Gamma_4^+ + \Gamma_5^+ \quad (8)$$

using the character orthogonality relations and where '*' denotes complex conjugation.

A general state in this three-band space can be expanded in the basis $\{|\phi_i\rangle\}$ of the $R_{4^-}$ representation as:

$$|\psi_{\mathbf{k}}\rangle = c_i(\mathbf{k})|\phi_{\mathbf{k}}^i\rangle, \quad (9)$$

adopting the Einstein summation convention. The energy expectation value is a scalar invariant, which is computed as:

$$\langle H \rangle = \langle \psi_{\mathbf{k}}|H|\psi_{\mathbf{k}}\rangle = c_i^*(\mathbf{k})c_j(\mathbf{k})H(\mathbf{k})_{ij}. \quad (10)$$

We seek combination of bilinears $c_i^* c_j$ transforming as the irreps above and take the Hermitian scalar product with functions of $\mathbf{k}$ and $\mathbf{B}$ with the same symmetry properties. From each term with $c_i^* c_j$ in this scalar product, it is easy to obtain the matrix elements of $H(\mathbf{k}, \mathbf{B})$. The energy scalar is written in this scalar product form:

$$\langle H \rangle = \sum_{\alpha, i} C_i^\alpha (q_i^\alpha)^* \cdot p_i^\alpha \quad (11)$$

where the sum runs over the irreps in the decomposition and $\{q_i\}$ and $\{p_i\}$ are the symmetry-adapted bases of the state coefficients and $\mathbf{k}$ and $\mathbf{B}$, respectively. The coupling constants $C_i^\alpha$ are parameters of the model.

To find the bases of bilinears in the wave coefficients transforming as the irreps above, we use the representation $\rho_{R_4^-}$ of the generators (omitting inversion for

---

**Fig. 5 Topological characterization via Wilson loops.** Wilson loops for the system in the Weyl semimetal phase (in blue) and in the cubic 3D $C_z$I phase (in red). Each $\theta$ eigenvalue of the WL is related to the WEC in terms of the $i$-th lattice parameter $|a_i|$ by $2\pi WEC_{n,i} \equiv |a_i|\theta_{n,i}$, with $|a_i| = |a|$ and $|a_i| = N|a|$ for the WS and the 3D CI respectively. **a, b** Individual photonic WECs: flow in the $k_x$ direction on a selected fixed $k_z$ plane, for each of the bands lying below the local gap. **c, d** Net photonic WEC flow, obtained by summing up the contribution of the individual WECs shown in the above panels: the net winding appears now clearly. Source data are provided as a Source Data file.

evaluated on a $k_y$ discretized closed loop $l$ crossing the BZ torus with

$$\hat{M}_{m,n}^{k_{y_i}, k_{y_{i+1}}} = \langle u_{m,k_{y_i}}|u_{n,k_{y_{i+1}}}\rangle. \quad (4)$$

Here $u_{m,\mathbf{k}}(\mathbf{r})$ represents the periodic part of the electromagnetic Bloch wavefunctions $\Psi_{m,\mathbf{k}}(\mathbf{r}) = \begin{pmatrix} \mathbf{E}_{m,\mathbf{k}}(\mathbf{r}) \\ \mathbf{H}_{m,\mathbf{k}}(\mathbf{r}) \end{pmatrix}$, for each mode of momentum $\mathbf{k}$ and band $m$, according to the relation $\Psi_{m,\mathbf{k}}(\mathbf{r}) = e^{-i\mathbf{k}\cdot\mathbf{r}}u_{m,\mathbf{k}}(\mathbf{r})$. In order to perform the correct equivalence between WCC and WEC, the scalar product needs to be weighted over the medium permittivity $\varepsilon$, permeability $\mu$ and bianisotropy $\chi$ tensors[52,53]:

$$\langle u|v \rangle = \int d^3\mathbf{r} u^\dagger \begin{pmatrix} \varepsilon & \chi \\ \chi^\dagger & \mu \end{pmatrix} v \quad (5)$$

As detailed in Ref. [40], in order to cure the effect of a possible arbitrary phase gained by the fields from numerical evaluation at different points on the $l$ path, we enforce periodic boundary conditions at the endpoints $k_{y_i}$ and $k_{y_i+b_y}$, where $b_y$ is the reciprocal $y$ lattice vector. Differently from 2D, we can avoid the singularity at $\omega = 0, \mathbf{k} = 0$[54,55] by looking only at planar cuts which do not include the $\Gamma$ point; this is sufficient for determining the Chern vector of our model. From the winding of the hybrid WEC in the BZ, the electromagnetic Chern number can be directly calculated as follows:

$$2\pi C_z^{EM} = \int_{-\pi}^{\pi} \sum_{n=1}^{\nu_{occ}} d\theta_{n,y}(k_x) \quad (6)$$

i.e. integrating along the $k_x$ closed path across the BZ and summing up the contribution of transverse $y$ WEC for the entire set of $\nu_{occ}$ bands lying below the local gap. Notice that all the definitions remain valid under cyclic permutations ($k_i, \theta_j, C_k$) of the Cartesian indexes $(ijk) = (xyz)$. Fig. 5 displays hybrid WEC for the 3D CI phase of Fig. 1f, comparing their individual to their total net contribution. We note that, for our particular non-bianisotropic ($\chi = 0$) system, it is possible to perform a simplification in the computation, decoupling electric and magnetic fields: under these conditions[56], the Chern number calculated in terms of just the electric ($C^E$) or just the magnetic field ($C^M$) is related to the total electromagnetic Chern number ($C^{EM}$) as:

$$C^{EM} = \frac{C^E + C^M}{2} = C^E = C^M. \quad (7)$$

**Analytical models.** In this section, we set up two symmetry-adapted models that describe: first, the threefold degeneracy at $\mathbf{R}$ and its splitting in $\mathbf{k}$ space when an external magnetic field $\mathbf{B}$ is applied, to give rise to a pair of Weyl points; second, the merging of the Weyl points by a lattice-commensurate modulation into a gapped topological phase.

brevity, as it is represented by the negative identity matrix):

$$\rho_{R_{\bar{4}}}(2_{001}) = \begin{pmatrix} -1 & 0 & 0 \\ 0 & -1 & 0 \\ 0 & 0 & 1 \end{pmatrix}, \rho_{R_{\bar{4}}}(3^{+}_{111}) = \begin{pmatrix} 0 & 1 & 0 \\ 0 & 0 & 1 \\ 1 & 0 & 0 \end{pmatrix},$$

$$\rho_{R_{\bar{4}}}(2_{110}) = \begin{pmatrix} 0 & 1 & 0 \\ 1 & 0 & 0 \\ 0 & 0 & -1 \end{pmatrix}, \tag{12}$$

where, for convenience, we have labeled the matrices by the rotation part of the symmetry element only. Note that these differ from the BCS data by a permutation of the basis, chosen to rearrange $H$ in a more convenient form. Since it is a non-symmorphic space group, some operations have fractional translations. In Seitz notation, these are:

$$\left\{ 2_{001} | \frac{1}{2}, \frac{1}{2}, 0 \right\}, \left\{ 2_{110} | \frac{1}{2}, \frac{1}{2}, 0 \right\}. \tag{13}$$

We then find functions of $\mathbf{k}$ and $\mathbf{B}$ with the same transformation properties, up to second order in the wave vector. In principle, the magnetic field could be strong. Therefore, the criterium to choose the maximum power of $\mathbf{B}$ that is included for each order in $\mathbf{k}$ is to exhaust all the possibilities in the irrep decomposition. This way, we ensure that all the couplings allowed by symmetry are included for a given order in the wave vector. Finally, we require $H(\mathbf{k}, \mathbf{B})$ to be Hermitian.

Following this procedure, we find that the most general expression for the energy operator is:

$$H = (a_0 k^2 + \alpha_0 B^2)\mathbb{1}_3 + i\delta_0\varepsilon_{jkl}B_l + b_0\begin{pmatrix} k_x^2 & 0 & 0 \\ 0 & k_y^2 & 0 \\ 0 & 0 & k_z^2 \end{pmatrix}$$

$$+ \beta_0\begin{pmatrix} B_x^2 & 0 & 0 \\ 0 & B_y^2 & 0 \\ 0 & 0 & B_z^2 \end{pmatrix} + c_0\begin{pmatrix} 0 & k_xk_y & k_xk_z \\ k_xk_y & 0 & k_yk_z \\ k_xk_z & k_yk_z & 0 \end{pmatrix}$$

$$+ \gamma_0\begin{pmatrix} 0 & B_xB_y & B_xB_z \\ B_xB_y & 0 & B_yB_z \\ B_xB_z & B_yB_z & 0 \end{pmatrix}, \tag{14}$$

where $\mathbf{k} = (k_x, k_y, k_z)$ is the wave vector measured from the point $\mathbf{R}$ and we employ real coefficients (Latin when referring to $\mathbf{k}$ and Greek to $\mathbf{B}$).

One can check that the energy operator is invariant under the little-group symmetries as it verifies, for every operation $g = \{R|\mathbf{t}\}$:

$$\rho_{R_{\bar{4}}}(g)H\rho_{R_{\bar{4}}}(g)^{-1} = H(R\mathbf{k}, R\mathbf{B}). \tag{15}$$

We also have imposed that the model be invariant when both the system and the external magnetic field $\mathbf{B}$ are transformed by time reversal $\Theta$. We can express the TR operation as $\Theta = U\kappa$, where $U$ is a unitary matrix and $\kappa$ is the complex conjugation operator. Then, the TRS condition reads:

$$\Theta H(\mathbf{k}, \mathbf{B})\Theta^{-1} = UH^*(\mathbf{k}, \mathbf{B})U^{-1} = H(-\mathbf{k}, -\mathbf{B}) \tag{16}$$

where the unitary $3 \times 3$ matrix part has the simple form $U = \mathbb{1}_3$. Evaluating the model at the point $\mathbf{R}$, in the presence of TRS, allows us to fix some of the coefficients by comparing with the numerical simulations, as described in Fig. 6. This yields $b_0 \sim 2.9a_0$.

When a magnetic field is applied along one of the coordinate axes $\mathbf{B} = B_i\hat{\mathbf{x}}_i$, the energy dispersion of the three photonic modes along the line parallel to the field is:

$$\begin{cases} \omega_1 = (a_0 + b_0)k_i^2 + (\alpha_0 + \beta_0)B_i^2 \\ \omega_2 = a_0 k_i^2 + \alpha_0 B_i^2 - \delta_0 B_i \\ \omega_3 = a_0 k_i^2 + \alpha_0 B_i^2 + \delta_0 B_i. \end{cases} \tag{17}$$

This further fixes $\alpha_0 \sim -\beta_0 \sim 0$ and shows that the magnetic field fully lifts the threefold degeneracy. We see in Fig. 6(b) that the band curved upwards in energy will cross with one of the remaining two, giving rise to a Weyl point. The strength of the magnetic field tunes where this crossing happens along this line, according to the expression:

$$k_i^{\pm} = \pm\sqrt{\frac{|\delta_0 B_i|}{b_0}}. \tag{18}$$

The same happens in the opposite direction along the same line, hence the $\pm$ sign. This shows that a Weyl dipole appears along the line parametrized by $k_i$ and that the position of the nodes can be tuned by the magnetic field strength $B_i$.

*Coupling of Weyl points by supercell modulation.* Because the validity of the previous analysis is limited to the neighborhood of the point $\mathbf{R}$, we construct another model that expands directly around the Weyl points. In particular, we can fix the magnetic field to $\mathbf{B} = B_z\hat{\mathbf{z}}$ and tune the strength $B_z$ to create a pair of Weyl points at $\mathbf{K}_{1,2} = (1/2, 1/2, \pm 1/4)$. The Weyl nodes can then be coupled by a supercell modulation that doubles the real-space unit cell in the $\hat{\mathbf{z}}$ direction. This

---

**Table 2 Decomposition of the product of representations $D^{\star} \times D$. Each row labels the decomposition of the term-wise product, considering that $D = D_1 + D_5$ is derived from a space group irrep induced from the sum of small irreps $T_1 + T_5$, which is then restricted to the arms $K_{1,2}$. The terms in the reduction are small irreps of the little groups at $\Gamma$ and $X_3 = (0, 0, 1/2)$. The labels for the $X$ irreps are those from the point $X \equiv X_2 = \frac{\mathbf{b}_y}{2} = (0, 1/2, 0)$, since $X_3$ is in the star of $X$.**

| $D^{\star} \times D$ [a] | $\Gamma$ [b] | $X_3$ [c] |
|---|---|---|
| $D_1^* \times D_1$ | $\Gamma_1^+ + \Gamma_3^+ + \Gamma_4^-$ | $X_1$ |
| $D_1^* \times D_5$ | $\Gamma_4^+ + \Gamma_4^- + \Gamma_5^+ + \Gamma_5^-$ | $X_3 + X_4$ |
| $D_5^* \times D_1$ | $\Gamma_4^+ + \Gamma_4^- + \Gamma_5^+ + \Gamma_5^-$ | $X_3 + X_4$ |
| $D_5^* \times D_5$ | $\Gamma_1^+ + \Gamma_1^- + \Gamma_2^+ + \Gamma_2^-$ | |
| | $+2\Gamma_3^+ + 2\Gamma_3^- + \Gamma_4^+$ | $2X_1 + 2X_2$ |
| | $+\Gamma_4^- + \Gamma_5^+ + \Gamma_5^-$ | |

[a]Product of representations of the $G_W$ little group.
[b]Irreps of the little group at the Gamma point.
[c]Irreps of the little group at the point (0, 0, 1/2).

---

corresponds to the uniaxial $N_W = N = 2$ case in the main text, i.e. a $(1, 1, 2)$ supercell. A generalization to cubic $(N, N, N)$ supercells is straightforward, since each Cartesian component of the modulation can be turned on independently.

The compatibility relations from $\mathbf{R}$ into the $\mathbf{T} = (1/2, 1/2, u)$ line yield:

$$R_4^-(3) \rightarrow T_1(1) + T_5(2), \tag{19}$$

where the dimensions of the small irreps are in parentheses. The magnetic field splits the states of $T_5$ and one of them is degenerate with the $T_1$ state at $\mathbf{K}_{1,2}$. We set up a model that describes these six photonic modes and its coupling by a cell modulation commensurate with the lattice.

Both $\mathbf{K}_1$ and $\mathbf{K}_2$ belong to the same star and are related by inversion. The representation that acts on the six photonic states is obtained from the space group representation induced from the direct sum $T_1 + T_5$. We then restrict this representation only to the $\mathbf{K}_{1,2}$ arms and consider the elements which either leave $\mathbf{K}_i$ invariant or relate one to the other. The subspace of these two arms is invariant under all these elements, which form a group that we denote by $G_W$.

Let us call $D$ the representation of $G_W$ so obtained. $D$ is divided into blocks arising from the $T_1$ and $T_5$ irreps, hence we write $D = D_1 + D_5$. The direct product of the full space group irrep can be used to find the reduction of the product $D^* \times D$ into small irreps of the little groups at $\Gamma$ and $X_3 = \frac{\mathbf{b}_z}{2} = (0, 0, 1/2)$. The result is shown in Table 2. Note that the label for the $\mathbf{X}$ point differs from that in the main text (where it is called $\mathbf{X}_2$) and was chosen for consistency with the BCS notation.

We may divide the $6 \times 6$ energy operator matrix into $3 \times 3$ blocks. Then, the diagonal blocks are identified with the Weyl points and the off-diagonal ones with the modulation that couples the states at both nodes. In view of Table 2, we can also anticipate that the $\Gamma$ irreps will yield the elements of the diagonal blocks, and the $X_m$ ($m = 1, 2, 3, 4$) irreps the off-diagonal ones. This is consistent with lattice translations having non-trivial representation for the $X_m$ irreps, which means that they couple non-equivalent points in the BZ.

The matrices of $D$ that are needed to find the symmetry-adapted bases are the following (again, we omit the representation matrix of inversion for brevity):

$$\rho_{D_1}(2_{001}) = \begin{pmatrix} 1 & 0 \\ 0 & 1 \end{pmatrix}, \rho_{D_1}(m_{010}) = \begin{pmatrix} e^{-i\pi/4} & 0 \\ 0 & e^{i\pi/4} \end{pmatrix},$$

$$\rho_{D_1}(4^+_{001}) = \begin{pmatrix} e^{\frac{i\pi}{4}} & 0 \\ 0 & e^{-i\pi/4} \end{pmatrix} \tag{20}$$

and

$$\rho_{D_5}(2_{001}) = \begin{pmatrix} -\mathbb{1}_2 & 0 \\ 0 & -\mathbb{1}_2 \end{pmatrix},$$

$$\rho_{D_5}(m_{010}) = \begin{pmatrix} 0 & e^{-i3\pi/4} & 0 & 0 \\ e^{i\pi/4} & 0 & 0 & 0 \\ 0 & 0 & 0 & e^{-i\pi/4} \\ 0 & 0 & e^{i3\pi/4} & 0 \end{pmatrix},$$

$$\rho_{D_5}(4^+_{001}) = \begin{pmatrix} 0 & e^{-i3\pi/4} & 0 & 0 \\ e^{-i3\pi/4} & 0 & 0 & 0 \\ 0 & 0 & 0 & e^{-i\pi/4} \\ 0 & 0 & e^{-i\pi/4} & 0 \end{pmatrix}. \tag{21}$$

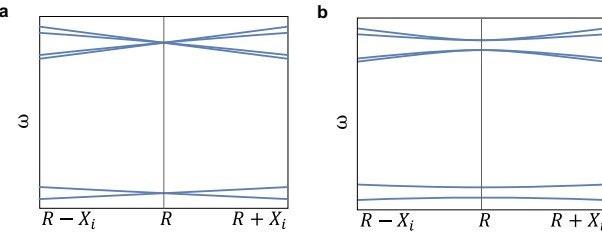

**Fig. 7 Effect of the $X_1$ modulation and Weyl points coupling.** From the analytical model, a four-fold degeneracy point is achieved at the folding condition $B_z = \frac{\delta}{\alpha - \beta}$, as shown in panel **a**. When a $X_1$ modulation is introduced, the Weyl dipole couples and a band-gap is opened, as showed in panel **b**. Parameters: $a = -7$, $b = -30$, $\alpha = 1$, $c = 1$, $d = 30$, $\gamma = 1$, $\beta = 0.45$, $\delta = 7.5$, $C_{1,2,3} = 1$, $p_1 = 7$, $q_1 = 0$.

Following the same procedure as in the Methods section, we obtain the matrix representation of the energy operator. As a last step, we impose TRS. The unitary part of the TR operation $\Theta = U\kappa$ is in this case

$$U = \begin{pmatrix} 0 & 1 \\ 1 & 0 \end{pmatrix} \otimes \begin{pmatrix} 1 & 0 & 0 \\ 0 & -1 & 0 \\ 0 & 0 & 1 \end{pmatrix}. \quad (22)$$

We are interested in modulations implemented by physically altering the dielectric structure of the crystal. Therefore, the modulation itself is considered to transform trivially under $\Theta$.

TRS forbids one of the couplings for each one of the $X_2$, $X_3$ and $X_4$ modulations. Furthermore, analyzing the effect of these three by numerically diagonalizing the $H$ matrix shows that only modulations transforming in the $X_1$ representation can gap out the Weyl points. For examples of modulations that do not open a gap, see SN 12.

The representation matrices used to obtain the $X_1$ modulation terms are the following:

$$\rho_{X_1}(2_{001}) = \begin{pmatrix} 1 & 0 \\ 0 & 1 \end{pmatrix}, \ \rho_{X_1}(m_{010}) = \begin{pmatrix} 0 & -1 \\ 1 & 0 \end{pmatrix},$$
$$\rho_{X_1}(4^+_{001}) = \begin{pmatrix} 0 & -1 \\ 1 & 0 \end{pmatrix}. \quad (23)$$

We present the expression along the T line of $H(k_z, B_z)$ with only the $X_1$ couplings included:

$$H(k_z, B_z) = \begin{pmatrix} W_+ & X_1 \\ X_1^\dagger & W_- \end{pmatrix}, \quad (24)$$

$$W_\pm = \begin{pmatrix} ak_z^2 \pm bk_z + \alpha B_z^2 & 0 & 0 \\ 0 & ck_z^2 \pm dk_z + \beta B_z^2 & B_z\delta \pm \gamma k_z B_z \\ 0 & B_z\delta \pm \gamma k_z B_z & ck_z^2 \pm dk_z + \beta B_z^2 \end{pmatrix}, \quad (25)$$

$$X_1(p_1, q_1) = \begin{pmatrix} C_1(p_1 - iq_1) & 0 & 0 \\ 0 & C_2 p_1 - iC_3 q_1 & 0 \\ 0 & 0 & -iC_2 q_1 + C_3 p_1 \end{pmatrix}, \quad (26)$$

where the coordinates $(p_1, q_1)$ transform as $X_1$ and parametrize the modulation strength, and $C_{1,2,3}$ are real coupling constants, while the rest of parameters are also real. The $k_z$ component is taken from the point where the Weyl points merge after the cell folding.

The effect of this modulation is visualized in Fig. 7. The band-gap opened via supercell modulation in the TRS broken system is shown to be a Chern gap in the main text, by numerical means.

From the analytical model, we also observe that, to exactly superimpose the Weyl points, we need to tune the magnetic field to the folding condition:

$$B_z = \frac{\delta}{\alpha - \beta}, \quad (27)$$

as can be seen by diagonalizing the matrix $H(0, B_z)$.

As stated before, this model addresses the case where $N = N_W = 2$. When $N_W = 2$ is fixed but $N = 2n$ with $n$ integer (see Fig. 3), the modulation belongs to point $(0, 0, 1/N)$ and must enter at order $n$ in $H$. Therefore, unless $N_W = N = 2$, the modulation will belong to the high-symmetry line $\Delta$. The expression for the modulation for every $N = N_W > 2$ is given in the SN 13.

*Example of cell modulation.* We use the projectors onto the $i$-th basis element in the space of the irrep $X_1$:

$$P_{ii} \propto \sum_{g \in G} X_1^*(g)_{ii} g \quad (28)$$

where $g$ runs over the little co-group at $\mathbf{X}_3 = (0, 0, 1/2)$ and we disregard any normalization factors. Applying these to an arbitrary function $f(z)$, we find:

$$X_1 : \begin{cases} P_{11}f \propto f(z) + f(-z) \\ P_{22}f \propto f(z) - f(-z) \end{cases} \quad (29)$$
$$X_2, X_3, X_4 : \ P_{ii}f = 0, \quad i = 1, 2.$$

Therefore, given functions of $z$ that under lattice translations obey $\mathbf{T}f = e^{i\mathbf{X}_3 \cdot \mathbf{T}}f$, those that provide a basis for this irrep are one even and one odd, respectively. This proves that a modulation of the radius of the rods $\Delta r = r(z) - r_0 = r_m \cos(2\pi z/N|a|)$ with $N = 2$ belongs to $X_1$. In particular, it is parametrized by $(p, q) = (p, 0)$ in the model given in Methods section. We also note that, when the modulation is cubic, i.e. $\Delta r = r(x, y, z) - r_0 = r_m[\cos(\pi x/|a|) + \cos(\pi y/|a|) + \cos(\pi z/|a|)]$, it is only the $z$ dependent part that is responsible for gapping the Weyl points generated in a $B_z$ field.

We also note that our derivation was performed employing Hermitian perturbations. However, via the introduction of non-Hermitian perturbations in the model, it could be possible to incorporate the effect of losses (or gain) in the system. Non-Hermitian terms usually lead to a spread of the Chern bands along the imaginary axis, which therefore transform into band regions in the complex plane. As long as the effect is limited enough not to lead to merging of the two bands, it is generally possible to separate the two bands by a line-gap. Indeed, as shown in Ref. [61], in presence of a line-gap, Chern insulators are stable with respect to non-Hermitian lossy effects.

## Data availability

Source data are provided with this paper. The data for Figs. 1–6 and Table 1 generated in this study are provided in the Supplementary Information/Source Data file. Source data are provided with this paper.

## Code availability

Code source data for the computation of photonic Wilson loops are available from the corresponding authors upon reasonable request. We are working on making the codes user friendly and we aim to present them as the object of a future publication devoted to the topological characterization of 3D photonic systems.

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

## Acknowledgements

The authors dedicate this work to the memory of their beloved colleague and friend, Prof. Alexey A. Soluyanov, who passed away on October 26, 2019. A.G.E., C.D. and M.B.P. acknowledge support from the Spanish Ministerio de Ciencia e Innovación (PID2019-109905GA-C2) and from Eusko Jaurlaritza (IT1164-19, KK-2019/00101 and KK-2021/00082). M.G.D., I.R. and M.G.V. acknowledge the Spanish Ministerio de Ciencia e Innovacion (grant PID2019-109905GB-C21). J.L.A. acknowledges support from the Spanish Ministerio de Ciencia e Innovación (PID2019-109905GA-C2). The work of B.B. is supported by the Air Force Office of Scientific Research under award number FA9550-21-1-0131. C.D. acknowledges financial support from the MICIU through the FPI PhD Fellowship CEX2018-000867-S-19-1. The work of J.L.M. has been supported by Spanish Science Ministry grant PGC2018-094626-BC21 (MCIU/AEI/FEDER, EU) and Basque Government grant IT979-16. A.G.E. and M. G. V. acknowledge funding from Programa Red Guipuzcoana de Ciencia, Tecnología e Innovación 2021 (Grant Nr. 2021-CIEN-000070-01. Gipuzkoa Next).

## Author contributions

A.G.E., M.G.V. and C.D. initiated the project. A.G.E., M.G.V., B.B. and J.L.M. outlined the work. A.G.E., M.G.V., C.D. M.G.D., J.L.M., I.R. and B.B. developed the theory. C.D., A.G.E., M.G.D., J.L.A. and M.B.P. performed the simulations. All the authors discussed and analyzed the results. C.D., M.G.V., A.G.E. and B.B. wrote the manuscript with input from all coauthors. A.G.E. and M.G.V. coordinated the project.

## Competing interests

The authors declare no competing interests.
