## [Peer Review File · Nature Communications]

REVIEWER COMMENTS

Reviewer #1 (Remarks to the Author):

In the work "Cubic 3D Chern photonic insulators with orientable large Chern insulator", Devescovi and co-authors propose a general approach to design 3D photonic Chern insulator and control both the direction and magnitude of Chern number vector via such approach. While Chern number is defined upon 2D manifold and Chern number vector for 3D structure is oriented along the magnetization axis are well-known facts, the novel and interesting aspect in this work is, in my opinion, the summation of section Chern number by band-folding the Brillouin zone, with such strategy employed, Chern numbers can be made arbitrarily large, or the requirement for large magnetic strength in the realistic application of topological photonics is relaxed. My specific comments and concerns are made in the following.

The authors claim throughout their paper they can realize the 3D CI in a weakly TRS broken environment, but in the numerical examples, the magnetic strength factor η is very large. For example, even for band-fold case $N=3$, we would expect a small value η , but $\eta/\epsilon=7.8/16$. Such claim will be more persuasive if they provide examples with smaller η and without compromising topological bandgap size.

Any realistic gyroelectric or gyromagnetic materials have dispersion and thus the loss is an unavoidable issue. Can the authors provide some discussions about the dissipation effect of the materials on the topological properties of their proposed structure?

It is not rigorous to say the surface states are unidirectional because they exist on 2D sheet. If the authors insist on such word, the accurate meaning of word "unidirectional" should be given. Besides, the equi-frequency surfaces are not disjoint since they live on 2D BZ so the surface dispersions are continuous and connected if they translate 2π in the k_y direction (Fig. 4c).

The large Chern number is achieved by a larger supercell and appropriate modulation of the elements in the supercell. Although the authors calculate the photonic Wilson loops and Chern numbers directly, it is also interesting to show the band structures supporting the gapless surface states for these large Chern number cases.

Overall, this paper is well written and provides potential implications toward the application of topological photonics, I would recommend the publication in Nature Communications after the authors provide more numerical evidences to support their claim and address questions above and from other referees.

Reviewer #2 (Remarks to the Author):

Comments on the manuscript entitled "Cubic 3D Chern photonic insulators with orientable large Chern vectors", authored by Dr Garcia-Etxarri and colleagues.

In this manuscript, based on the Brillouin zone folding and supercell modulation mechanism, the authors proposed a general strategy to design three-dimensional time-reversal-symmetry-breaking cubic photonic Chern insulators with orientable and arbitrarily large Chern vectors. Specifically, they showed that the direction of the Chern vector can be simply tuned by changing the orientation of the applied magnetic field. An elegant combination of the group-theoretical method and photonic analog of

Wilson loops was used to characterize the photonic band structure and corresponding topology.

Overall speaking, I think the physics proposed in this manuscript is very interesting, and it may attract the broad interest of both photonic and condensed matter communities. The manuscript is presented in an easily comprehensible way and the derivation is quite explicit with the necessary details provided in the Methods part and Supplementary Information. Therefore, I am inclined to recommend it to be published in Nature Communications. Nevertheless, I still have some concerns that need to be addressed before my recommendation.

(1) The authors wrote that the surfaces 'Fermi' loops can be split into two disjoint sections, but according to the equifrequency loops in Fig. 4c, it seems that the blue and red sections are touching at some points on the surface Brillouin zone boundary of $k_y = 0.5$. Is there anything special about these touching points? Are they just accidental?

(2) It is well known that Weyl semimetals are characterized by Fermi arc surface states. What is the fate or evolution of the Fermi arc states after Brillouin zone folding and supercell modulation? Can the chiral and unidirectional surface states be understood from the remnant of the Fermi arc states?

(3) In this manuscript, Weyl points are introduced by breaking time-reversal symmetry with the application of an external magnetic field. What if one considers another route of generating Weyl points by break inversion symmetry, say, by some specific geometrical designs of the photonic system? If time-reversal symmetry is preserved in this process, is it possible to induce a three-dimensional time-reversal-invariant topological insulator phase through the folding and gap-opening mechanism?

(4) Apart from the No. 224 space group, can the strategy presented in this manuscript be generalized to other space groups?

(5) Some typos and grammatical errors should be corrected, for example, in the caption of Fig. 4, the label of 4f is wrongly written as 4e. In the seventh line, left column of page 2, 'a engineering...' should be corrected as 'an engineering...'.

Reviewer #3 (Remarks to the Author):

The paper by Devescovi et. al. proposed a new way to design a 3D Chern photonic insulator, which does not need large magnetization or magnetic field and which can realize higher Chern integers. A basic ingredient of their proposal is a pair of Weyl nodes that appears around a high symmetric point in a photonic band of a cubic cell of four dielectric rods under the magnetic field. By replicating this original unit cell in a supercell, the authors design many pairs of Weyl nodes overlap with one another, such that the Weyl node with positive charge and the Weyl node with negative charge are gapped out with each other. By doing so, the authors realize 3D Chern photonic insulator with reduced magnetization, and with large Chern numbers. The authors further demonstrate the presence of chiral surface states in these 3D Chern photonic insulators. The paper is based on analytical argument and numerical calculations. I found their argument sounds reasonable and their statement seems to be feasible. In fact, the 'method to design 3D topological photonic crystal where the Chern vector of any magnitude, sign or direction can be implemented at will' is very important for a future realization of 'photonic on-chip communications with higher channel capacity' and I believe that in this respect, the paper will attract a lot of interests in a field of photonics and topological photonics. Therefore, I recommend the publication of the paper after the authors take into account following comments or questions.

1) When the authors make a supercell by replicating the original cell with the four rods, they used

either a cubic

supercell of dimensions (N,N,N) or a uniaxial supercell of size N . The former supercell is drawn in Fig. 1d, 1e, while the latter supercell are drawn in Fig. 2b, 2d, and Fig. 3. Apparently, 3D periodic lattice of the uniaxial supercell looks very similar to 3D periodic lattice of the original cell, while I believe that there must be some specific model parameters that differentiate these two situations. It would be helpful, if the authors explain what relevant material parameters will distinguish these two situations.

2) When the authors introduce supercell modulation to gap out the Weyl nodes, it is important to make the supercell structure to be commensurate to the distance between the two Weyl nodes in the original cell. The paper becomes comprehensive if the authors explain specially what would happen if they are not commensurate or far from the commensurate case.

3) The authors assume that the field are along either x , y or z in the cubic lattice. How will the pair of the Weyl nodes deviate from the high symmetric line in the momentum space, when the field is deviated from the symmetric axis ?

When a pair of the Weyl nodes deviate from the high symmetric line, how can the authors create the 3D chern insulators ?

4) Related to 3), how can the authors create those chern photonic crystals whose the Chern vector is not along the high symmetric line but along an arbitrary direction ?

Reviewer #4 (Remarks to the Author):

In the manuscript entitled "Cubic 3D Chern photonic insulators with orientable large Chern vectors," the authors Devescovi et al propose a 3D photonic crystal structure that breaks time-reversal symmetry (based on magneto-optical elements) that exhibits arbitrarily high sectional Chern numbers. These are closely related to photonic crystals with Weyl points in the band structure, and indeed the authors base their analysis on the creation (and then annihilation) of Weyl points (pairs of Weyl points actually – "Weyl dipoles"). Weyl points are can be interpreted as topological transition points in the Brillouin zone, where the Chern number changes as a function of a perpendicular wavevector. Via the use of Brillouin zone folding and ensuing perturbations, the authors demonstrate how one way manipulate the Weyl points to achieve arbitrarily high Chern numbers. The results seem to me to be thorough and correct.

That said, I can't recommend this work for Nature Communications for several reasons, mainly related to the experimental feasibility and device implications of their results. First, the dielectric parameters used are very high for optical materials, and are likely inaccessible except in the microwave regime (i.e., the type of demonstration experiments of Refs. [4] and [6]). Nano or microfabrication of materials with dielectric constants ~ 10 , let alone magnetic responses on that order has never been accomplished and seems to me far-fetched.

Moreover, the authors talk about how the multiple surface states could be used for achieving higher channel capacity. I can imagine how this could be true for Ref. [28] for 1D channels embedded in 2D, but a full 2D surface state embedded in 3D is not a "wire" that can carry information. Setting aside the point that this class of structures probably can't be realized for sub-cm scale wavelengths, I also don't see how using zone folding could give rise to higher channel capacity, even if there are nominally more surface states corresponding to higher Chern numbers.

Given the extreme difficulty of fabricating topologically non-trivial photonic crystals in 3D (especially with time-reversal symmetry breaking), my feeling is that a contribution more worthy of publication in Nature Communications would be one that constrains itself to the hard realities of current (or near-future) technology, and finds innovative solutions around these constraints.

Cubic 3D Chern photonic insulators with orientable large Chern vectors

Point-by-point response to the reviewers' comments

Chiara Devescovi,^{1,*} Mikel García-Díez,^{2,1} Iñigo Robredo,^{3,1} María Blanco de Paz,¹ Jon Lasa-Alonso,^{4,5}
Barry Bradlyn,⁶ Juan L. Mañes,⁷ Maia G. Vergniory,^{1,8,9,†} and Aitzol García-Etxarri^{1,8,‡}

¹*Donostia International Physics Center, 20018 Donostia-San Sebastián, Spain*

²*Physics Department, University of the Basque Country (UPV-EHU), Bilbao, Spain*

³*University of the Basque Country (UPV-EHU), Bilbao, Spain*

⁴*Centro de Física de Materiales, Paseo Manuel de Lardizabal 5, 20018 Donostia-San Sebastian, Spain*

⁵*Donostia International Physics Center, Paseo Manuel de Lardizabal 4, 20018 Donostia-San Sebastian, Spain*

⁶*Department of Physics and Institute for Condensed Matter Theory,
University of Illinois at Urbana-Champaign, Urbana, IL, 61801-3080, USA*

⁷*Physics Department, University of the Basque Country (UPV/EHU), Bilbao, Spain*

⁸*IKERBASQUE, Basque Foundation for Science, Maria Diaz de Haro 3, 48013 Bilbao, Spain*

⁹*Max Planck Institute for Chemical Physics of Solids, Dresden D-01187, Germany*

(Dated: September 22, 2021)

Dear Editors and Reviewers,

We thank the Editor for managing our manuscript and the Referees for the time they devoted to reviewing it. We believe that their input has allowed us to improve the quality of our manuscript substantially.

In the following, we respond point by point to all of their questions and concerns, indicating the changes we implemented in our article accordingly. Our point-by-point responses are highlighted in blue and are provided together with the original comments reproduced verbatim.

REVIEWER #1

In the work “Cubic 3D Chern photonic insulators with orientable large Chern insulator”, Devescovi and co-authors propose a general approach to design 3D photonic Chern insulator and control both the direction and magnitude of Chern number vector via such approach. While Chern number is defined upon 2D manifold and Chern number vector for 3D structure is oriented along the magnetization axis are well-known facts, the novel and interesting aspect in this work is, in my opinion, the summation of section Chern number by band-folding the Brillouin zone, with such strategy employed, Chern numbers can be made arbitrarily large, or the requirement for large magnetic strength in the realistic application of topological photonics is relaxed. My specific comments and concerns are made in the following.

The authors claim throughout their paper they can realize the 3D CI in a weakly TRS broken environment, but in the numerical examples, the magnetic strength factor η is very large. For example, even for band-fold case $N = 3$, we would expect a small value η , but $\eta/\epsilon = 7.8/16$. Such claim will be more persuasive if they provide examples with smaller η and without compromising topological band-gap size.

Regarding the possibility of further reducing the magnetic strength factor, we follow the Referee's suggestion and provide in Fig. R1 some examples with larger $N = N_W$. In these examples we have increased and optimized the modulation intensity in order to partially compensate for the band-gap decrease. We believe that these new results are now more persuasive than the ones included in the original manuscript. For instance, the case where $N = N_W = 7$ achieves an $\eta/\epsilon = 1.6/16$ which constitutes a significant reduction in the magnetic bias.

However, note that one cannot iterate this process down to zero magnetic field. Indeed, as the referee observes, there is a decrease in band-gap as $N = N_W$ grows, which is unavoidable: a decrease in the magnetic field necessarily leads to a compromise on the resulting gap. To visualize this, we also created a table (Tab. R1) of the band-gaps resulting from decreasing magnetization conditions in uniaxial supercells

Figure R1: Reducing the magnetic strength factor by increasing the supercell size N and the modulation intensity. The band-gap undergoes a reduction in size.

where the modulation intensity is chosen to maximize the band-gap at growing N . As it can be seen, the band-gap diminishes as $N = N_W$ grows, confirming the referee's observation. Indeed, in the limit of very large N , there is no splitting of Weyl points and thus no TRS broken gap can clearly be opened, $\lim_{N \rightarrow \infty} f_g = 0$.

$N = N_W$	5	6	7
η	3.0	2.1	1.6
$f_g/f_m(\%)$	1.5	1.4	1.0
$f_g(f a /c)$	0.005	0.004	0.003

Table R1: Reduction of the magnetic bias η and associated gap-to-midgap ratio $f_g/f_m(\%)$. Values are optimized via the modulation parameters, yet the band-gap reduction is unavoidable.

Building on this discussion, we now incorporate these new observations, Figure R1 and Table R1 to the SI section A.2. entitled "Minimizing the magnetic bias via multi-fold supercells".

We would also like to note that in the paper TRS breaking was implemented by using gyro-electric materials in an external magnetic field. However this TRS breaking can be equivalently achieved by exploiting the internal remnant magnetization of ferri-magnetic materials [1, 2]. In this case, it could be possible to obtain a large displacement of the Weyl points using currently available magnetic dielectrics. For example, by employing ferri-magnetic materials such as YIG [1, 2], it is possible to achieve values of the off-diagonal component of the gyro-magnetic permeability tensor of the order of ~ 12 , in the microwave regime. This is enough to displace the Weyl points up to half of the BZ (which we considered 'maximal' displacement). As a proof of fact, we show here the Weyl point band-structure for our geometry but using the experimentally tested gyro-magnetic parameters of Ref. [1, 2] where $\varepsilon = 15$, $\mu = 14$ and $\rho = 12.4$.

Here μ and ρ constitute the diagonal and off-diagonal component of the gyro-magnetic permeability tensor described as:

$$\mu_{\rho_z} = \begin{pmatrix} \mu_{\perp} & i\rho_z & 0 \\ -i\rho_z & \mu_{\perp} & 0 \\ 0 & 0 & \mu \end{pmatrix} \quad (1)$$

with $\mu_{\perp} = \sqrt{\mu^2 + \rho_z^2}$. The permittivity tensor is instead is described by a diagonal (isotropic) permittivity tensor, $\varepsilon_{TRS} = \varepsilon \mathbb{1}_3$.

Figure R2: Weyl point band-structure for our geometry realized using the 'gyro-magnetic' parameters of [1, 2] where $\varepsilon = 15$, $\mu = 14$ and $\rho = 12.4$.

Any realistic gyroelectric or gyromagnetic materials have dispersion and thus the loss is an unavoidable issue. Can the authors provide some discussions about the dissipation effect of the materials on the topological properties of their proposed structure?

The effect of losses (or gain) can be incorporated in the Hamiltonian via the introduction of non-Hermitian perturbations in the system. Non Hermitian terms lead to a spread of the Chern bands along the imaginary axis, which therefore transform into 'band regions' in the complex plane. As long as the effect is limited enough not to lead to merging of the two bands, it is generally possible to separate the two bands by a line-gap. In presence of a line-gap, as shown in Ref.[3], Chern insulators are stable with respect to non-Hermitian lossy effects.

Since we agree with the referee that this point is relevant, we included a line discussing it in the Methods section B.

It is not rigorous to say the surface states are unidirectional because they exist on 2D sheet. If the authors insist on such word, the accurate meaning of word "unidirectional" should be given. Besides, the equi-frequency surfaces are not disjoint since they live on 2D BZ so the surface dispersions are continuous and connected if they translate 2π in the k_y direction (Fig. 4c).

As suggested by the referee, we clarify the concept of unidirectionality that we adopt and incorporate it in the main document as well. Surface states are unidirectional in the following sense: The component of the group velocity (or Poynting vector) normal to the magnetization direction has a well defined sign

i.e. surface states cannot back-scatter along this specific direction. This component will be later denoted as conserved component. We define as 'chiral partners', the pair of surface states living on the opposite sides of the slab, moving with opposite component of the group velocity which is normal to magnetization axis.

Regarding the statement about disjoint equi-frequency surfaces, as well, we clarified and corrected the concept in the text. Specifically, we state that the equi-frequency loops can be separated in real space, i.e. we can associate those with positive group velocity component normal to the magnetization to a slab surface and those with negative one to the other slab surface.

The large Chern number is achieved by a larger supercell and appropriate modulation of the elements in the supercell. Although the authors calculate the photonic Wilson loops and Chern numbers directly, it is also interesting to show the band structures supporting the gapless surface states for these large Chern number cases.

We thank the referee for this suggestion. Of course, as an extra investigation, it would be interesting to look at the energy dispersion of the edge states. Unfortunately, computing the edge spectra for gapless surface states at large Chern numbers requires the use of large supercells. For instance, the calculation of the edge states for Chern number 1 (Fig. 4 in the main text), required of 300Gb of RAM memory and weeks of running time per k-point. Increasing the Chern number implies the calculation of larger supercells, meaning that the computational requirements of the calculations are much higher than those for $C = 1$. Although we tried, the simulations are numerically too challenging for our available computational resources. However, we would like to note that by bulk the boundary correspondence it is already possible to infer their presence at the surface, as many as what the Chern number dictates. Chern numbers are shown in Fig. 3 of the main text.

Overall, this paper is well written and provides potential implications toward the application of topological photonics, I would recommend the publication in Nature Communications after the authors provide more numerical evidences to support their claim and address questions above and from other referees.

REVIEWER 2

Comments on the manuscript entitled "Cubic 3D Chern photonic insulators with orientable large Chern vectors", authored by Dr Garcia-Etxarri and colleagues. In this manuscript, based on the Brillouin zone folding and supercell modulation mechanism, the authors proposed a general strategy to design three-dimensional time-reversal-symmetry-breaking cubic photonic Chern insulators with orientable and arbitrarily large Chern vectors. Specifically, they showed that the direction of the Chern vector can be simply tuned by changing the orientation of the applied magnetic field. An elegant combination of the group-theoretical method and photonic analog of Wilson loops was used to characterize the photonic band structure and corresponding topology.

Overall speaking, I think the physics proposed in this manuscript is very interesting, and it may attract the broad interest of both photonic and condensed matter communities. The manuscript is presented in an easily comprehensible way and the derivation is quite explicit with the necessary details provided in the Methods part and Supplementary Information. Therefore, I am inclined to recommend it to be published in Nature Communications. Nevertheless, I still have some concerns that need to be addressed before my recommendation.

(1) The authors wrote that the surfaces 'Fermi' loops can be split into two disjoint sections, but according to the equifrequency loops in Fig. 4c, it seems that the blue and red sections are touching at some points on the surface Brillouin zone boundary of $k_y = 0.5$. Is there anything special about these touching points? Are they just accidental?

Regarding the presence of touching points in the Fermi loops pointed out by the referee, we clarify here the concept of disjoint equi-frequency surfaces and incorporate the explanation in the text. Specifically, we state that the equi-frequency loops can be split in real space, i.e. we can associate different real space

surfaces of the topological slab to different chiral partners, as indicated by blue and red colors in Fig. 4c in the main text. Because of this, the presence of touching points between the surface states dispersion of different chiral partners, is perfectly tolerated. Indeed, these crossings occur between states that reside on opposite sides of the sample and are physically separated in real space by the bulk. Therefore they cannot gap out, up to exponentially small finite size effects, and are protected by the spatial separation separation of chiral partners on opposite surfaces. In conclusion, these touching points in the Brillouin zone mentioned are purely accidental and can be considered crossing points just if we neglect the spatial localization of the crossing states.

(2) It is well known that Weyl semimetals are characterized by Fermi arc surface states. What is the fate or evolution of the Fermi arc states after Brillouin zone folding and supercell modulation? Can the chiral and unidirectional surface states be understood from the remnant of the Fermi arc states?

As proposed by the referee, we investigate the fate of the Fermi arcs and describe their evolution during the supercell folding and modulating process. Our results can be summarized by Fig. R3, for an uniaxial system with $N = N_W = 2$. Panel A shows the Fermi arcs of the Weyl semimetallic phase. Opposite charged Weyl points are located at $(k_y^\pm, k_z^\pm) = (\pm\frac{\pi}{2}, \pi)$, i.e. midway along the **SR** line. After supercell folding, the Weyl points are superimposed and their Fermi arcs display a folding as well, as in panel B. Finally, by turning on the supercell modulation, a gap is opened in the BZ and the Fermi arcs sharply evolve into disjoint Fermi loops of the 3D CI system, as in panel C. Therefore, the Fermi arcs of the Weyl semimetal are transformed into the Fermi loops of the 3D CI system as result of the supercell folding and modulation.

Figure R3: Evolution of the Fermi arcs (a) after Brillouin zone folding (b) and supercell modulation (c).

In addition, in order to show a smoother evolution of Fermi arcs into loops, we also compute them for a system where we have reversed the order of TRS symmetry breaking and supercell modulation. Thus, in this new set of calculations, we first spatially modulate our structures and then we apply the magnetic field adiabatically increasing its value until the Weyl points annihilate. In this case, as shown in Fig. R4, Fermi points appear in the modulated system as small TRS breaking is introduced (panel A), they gradually move far apart as the magnetic bias is increased with their Fermi arcs getting longer (panel B), up to final annihilation in panel C. We add these new Figures to the SI where we create a section dedicated to Fermi arcs, section B.4. entitled "Evolution of WS 'Fermi' arcs into 3D CI 'Fermi' loops".

(3) In this manuscript, Weyl points are introduced by breaking time-reversal symmetry with the application of an external magnetic field. What if one considers another route of generating Weyl points by break inversion symmetry,

Figure R4: Evolution of the Fermi arcs reversing the order of TRS breaking and modulation. The starting setup is a supercell modulated system with small magnetic field (a). As they field is increased the Weyl point separations grows (b). Finally Weyl points annihilate and Fermi arcs evolve into Fermi loops (c).

say, by some specific geometrical designs of the photonic system? If time-reversal symmetry is preserved in this process, is it possible to induce a three-dimensional time-reversal-invariant topological insulator phase through the folding and gap-opening mechanism?

The argument of gap opening by folding and supercell modulation is very general, and can be applied as long as the constraint of commensurability between the Weyl displacement and the supercell size is satisfied. However, annihilating Weyl points in inversion breaking (chiral) photonic crystals might not lead necessarily to a topological gap. As observed by Refs. [4, 5], annihilating Weyl points in TRS preserving fermionic systems can result in a 3D Quantum Spin Hall phase. However, since we are dealing with a photonic (bosonic) system, we cannot rely on the protections of Kramers Theorem like in the conventional fermionic QSH. The effect of our supercell modulation strategy might therefore lead only to a 3D trivial gapped phase. Nevertheless, if properly choosing some combination of time-reversal and crystalline symmetries, it might be possible to generate some proper pseudo-spin invariant [6–8], leading to the formation of a 3D pseudo-Z2 phase. We are currently investigating this line of research. Nevertheless, we believe that those results fall outside the scope of this manuscript and will be eventually presented as a separate work.

(4) Apart from the No. 224 space group, can the strategy presented in this manuscript be generalized to other space groups?

To develop the strategy proposed in our paper, we employed space group No. 224 since it was an ideal platform to obtain TRS broken Weyl points with a tunable Weyl dipole direction. Indeed the system displays a single threefold degeneracy at \mathbf{R} which can be easily split into a Weyl dipole via magnetization. However, the supercell modulation mechanism developed does not rely on any specific symmetry of space group No. 224. Therefore, any other crystal structure exhibiting a pair of TRS Weyl points could be perfectly suited to obtain a 3D CI phase. The only fundamental constraint is that the chosen supercell generates commensurate folding in the Weyl dipole direction, in order to lead to a superposition and annihilation of the opposite charged Weyl points.

(5) Some typos and grammatical errors should be corrected, for example, in the caption of Fig. 4, the label of 4f is wrongly written as 4e. In the seventh line, left column of page 2, ‘a engineering...’ should be corrected as ‘an engineering...’.

As suggested, we correct these typos in the main text.

REVIEWER 3

The paper by Devescovi et. al. proposed a new way to design a 3D chern photonic insulator, which does not need large magnetization or magnetic field and which can realize higher Chern integers. A basic ingredient of their proposal is a pair of Weyl nodes that appears around a high symmetric point in a photonic band of a cubic cell of four dielectric rods under the magnetic field. By replicating this original unit cell in a supercell, the authors design many pairs of Weyl nodes overlap with one another, such that the Weyl node with positive charge and the Weyl node with negative charge are gapped out with each other. By doing so, the authors realize 3D Chern photonic insulator with reduced magnetization, and with large Chern numbers. The authors further demonstrate the presence of chiral surface states in these 3D Chern photonic insulators. The paper is based on analytical argument and numerical calculations. I found their argument sounds reasonable and their statement seems to be feasible. In fact, the ‘method to design 3D topological photonic crystal where the Chern vector of any magnitude, sign or direction can be implemented at will’ is very important for a future realization of ‘photonic on-chip communications with higher channel capacity’ and I believe that in this respect, the paper will attract a lot of interests in a field of photonics and topological photonics. Therefore, I recommend the publication of the paper after the authors take into account following comments or questions.

1) When the authors make a supercell by replicating the original cell with the four rods, they used either a cubic supercell of dimensions (N,N,N) or a uniaxial supercell of size N. The former supercell is drawn in Fig. 1d, 1e, while the latter supercell are drawn in Fig. 2b, 2d, and Fig. 3. Apparently, 3D periodic lattice of the uniaxial supercell looks very similar to 3D periodic lattice of the original cell, while I believe that there must be some specific model parameters that differentiate these two situations. It would be helpful, if the authors explain what relevant material parameters will distinguish these two situations.

We thank the referee for this question. We tried to explain these differences in the modulation strategies in the main text, but we will expand and revise our discussion to make it more clear:

Graphically, the supercell modulation is visualized by employing a scale of colors for the dielectric structure plots: for example, as shown in Fig. 1e, a colorbar is associated to the local radius of the cylinders. Note that in all our cases, the modulation applied is very small, $r_m \ll r_0$, which may result in a very subtle graphical difference between between the original lattice and the modulated one. Therefore, in order to better visualize the presence of the modulation, we updated all the figures of the paper, graphically amplifying the supercell modulation and adopting new colorbars.

In addition, we remark that uniaxial and cubic supercell modulated crystals are different in the following sense. In both cases the radius of the cylinders is locally varied, by locally changing the radius of the spheres in the covering approximation. However, the local change from the original r_0 radius to new local one $r(x, y, z)$ is performed according to the relation $\Delta r(x, y, z) = r(x, y, z) - r_0 = r_m \cos(2\pi x_i/N|a|)$, for the uniaxial case, while according to $\Delta r(x, y, z) = r(x, y, z) - r_0 = r_m[\cos(2\pi x/N|a|) + \cos(2\pi y/N|a|) + \cos(2\pi z/N|a|)]$, for the cubic case. More formally, one could introduce a vector of model parameters $\delta = (\delta_x, \delta_y, \delta_z)$ that differentiates the two situations as follows: $\Delta r(x, y, z) = r(x, y, z) - r_0 = r_m[\delta_x \cos(2\pi x/N|a|) + \delta_y \cos(2\pi y/N|a|) + \delta_z \cos(2\pi z/N|a|)]$ where $\delta = (1, 1, 1)$ for the cubic case where all components are modulated and $\delta = (0, 0, 1)$ for uniaxial modulation along \hat{z} .

This means that for a uniaxial supercell the modulation is performed along a single Cartesian axis, while for a cubic one the structure is modulated along all the Cartesian directions. To better differentiate the cases, here we report a plot (Fig. R5) where the modulation parameter has been largely amplified $r_m \sim r_0/5$ and where we compare cubic and uniaxial supercells, on a (222) lattice so that one could better follow the 3D periodicity.

In order to improve the explanation of these aspects of our work, we included a new section in the Supplementary Information entitled "Details on the different modulation strategies" with these observations and Figure R5.

2) When the authors introduce supercell modulation to gap out the Weyl nodes, it is important to make the supercell structure to be commensurate to the distance between the two Weyl nodes in the original cell. The paper becomes

Figure R5: Visualizing the supercell modulation when the model parameters are largely amplified $r_m \sim r_0/5$. Cubic (b) and uniaxial case (c). For the uniaxial case, we replicate laterally 4 unit cells, in order to better compare the 3D periodic structure with the cubic one.

comprehensive if the authors explain specially what would happen if they are not commensurate or far from the commensurate case.

As noted by the referee, the argument of gap opening by folding and modulation requires the supercell to be commensurate with the distance between the Weyl points. However, as shown in SI (section A.1 of the original submission), moderate deviations from the exact fine tuned conditions are tolerated. As it can be seen in Fig. R6 (Figure S1 in the SI entitled "Robustness of the topological gap" of the original submission), the gap opening occurs also when there is a moderate mismatch in reciprocal lattice vectors. For the specific case displayed, even when the magnetic fields deviates by one part in 16 from its optimal value, the resulting gap is still 75% of its maximal size. We interpreted this tolerance as an effect of the applied supercell modulation, which slightly moves together and then couples the Weyl points. As we move farther away from the commensurate condition, the band-gap gradually diminished, down to zero. Indeed, as we largely deviate from the commensurate case, the Weyl points cannot be superimposed by band folding and therefore cannot be coupled by the supercell modulation.

Figure R6: Weyl point annihilation occurring with tolerance over folding lattice vector and Weyl point location mismatch. (2, 2, 2) supercell with no supercell modulation (a) and after modulation (b).

3) The authors assume that the field are along either x, y or z in the cubic lattice. How will the pair of the Weyl nodes deviate from the high symmetric line in the momentum space, when the field is deviated from the symmetric axis ? When a pair of the Weyl nodes deviate from the high symmetric line, how can the authors create the 3D chern insulators ?

The strategy we developed to obtain 3D Chern insulators still applies when the magnetic field deviates from the x, y, z symmetry lines, as long as the Weyl dipole is oriented along an integer linear combination of the lattice vectors. In such cases, a proper auxiliary supercell can be constructed that is commensurate with the Weyl vectors. For example, consider applying the magnetic field along (110). Since the starting point is a cubic lattice, we expect the Weyl points to split along the same direction, as in Fig. R7; this relies on the fact that the system is a cubic lattice with O_h point group, and thus cyclic trifold symmetry in x, y, z .

Figure R7: Orienting the Weyl dipole along an integer linear combination of the Cartesian components. Splitting occurs along the \mathbf{ZRZ}' line where $\mathbf{Z}' = \mathbf{Z} - \mathbf{b}_y - \mathbf{b}_x$

For a (110) Weyl dipole orientation, the proper auxiliary supercell should have the following structure (N_x, N_y, N_z) with $N_x = N_y = N = N_W$ chosen to be commensurate with the Weyl dipole separation. For example, assume splitting the Weyl points at $N_W = 4$ with a field oriented along (110): by simple folding arguments, a band-gap can be opened via folding and modulating along (110) on a commensurate $N_x = N_y = N = 4$ supercell (either cubic with $N_z = N$, or anisotropic with $N_z \neq N$).

4) Related to 3), how can the authors create those Chern photonic crystals whose the Chern vector is not along the high symmetric line but along an arbitrary direction ?

As explained above, this goal can be achieved by combining the tilting of the Weyl dipole with a proper anisotropic commensurate integer supercell. This way it is possible to obtain a Chern vector whose direction is not strictly a Cartesian direction provided being an integer linear combination of them. In this case, the Weyl dipole should have the same orientation as the the Chern vector. For the example discussed previously, we can expect the resulting system to have a non-zero Chern vector along (110). We include these reasonings to the SI section A.6 entitled "Weyl dipole tilted from the Cartesian directions".

REVIEWER 4

In the manuscript entitled "Cubic 3D Chern photonic insulators with orientable large Chern vectors," the authors Devescovi et al propose a 3D photonic crystal structure that breaks time-reversal symmetry (based on magneto-optical elements) that exhibits arbitrarily high sectional Chern numbers. These are closely related to photonic crystals with Weyl points in the band structure, and indeed the authors base their analysis on the creation (and then annihilation) of Weyl points (pairs of Weyl points actually – "Weyl dipoles"). Weyl points are can be interpreted as topological transition points in the Brillouin zone, where the Chern number changes as a function of a perpendicular wavevector. Via the use of Brillouin zone folding and ensuing perturbations, the authors demonstrate how one way manipulate the Weyl points to achieve arbitrarily high Chern numbers. The results seem to me to be thorough and correct.

That said, I can't recommend this work for Nature Communications for several reasons, mainly related to the experimental feasibility and device implications of their results.

First, the dielectric parameters used are very high for optical materials, and are likely inaccessible except in the microwave regime (i.e., the type of demonstration experiments of Refs. [4] and [6]). Nano or microfabrication of materials with dielectric constants 10, let alone magnetic responses on that order has never been accomplished and seems to me far-fetched.

Regarding the first concern posed by the referee about experimental feasibility we highlight several points. The large dielectric parameters employed for our demonstration strategy can be largely reduced as explained in paragraph '3D CI at reduced magnetization' of the Results section, minimizing the magnetization requirements via the use of a larger modulation intensity combined with larger N supercells. This may allow experimental realization also via dielectric materials for which the magnetic response are not high. For example, as shown in Fig. 2 of the main text, by the use of a larger supercells it is already possible to halve the gyrotropic parameters from their maximal value at the same Chern number. The value of the gyrotropic parameters can be further decreased, as shown in our revised SI section A.2, provided compromising on the size of the topological band-gap. Nevertheless we aim to stress that our results are not necessarily intended to be realized strictly in the optical regime, where currently available materials are known to display a weak magnetic response. In the microwave regime for example, photonic crystal fabricated both in 2D [1] and 3D [9] have been shown to display dielectric values comparable to ours. In particular, in an experimental work published few days after our submission [9], annihilation of Weyl points was observed at large dielectric parameters via 3D YIG structures.

We also intend to emphasize that the strategy we devised in our paper is material agnostic, and can be easily adapted to any to-be-discovered experimental platform. In that sense, our work provides a roadmap to future experimental exploration of topological photonic crystals by showing how to reduce the needed magnetic response.

Finally, regarding the aspect of microfabrication, we similarly want to stress that the results have been given in a scale invariant setup (note our normalization with respect to the length parameters employed). Therefore, even if crystal geometry design and modulation might be currently challenging at the nano- or micro-scale, our proof of principle can be applied to photonic systems on a larger scale, where fabrication is feasible.

Moreover, the authors talk about how the multiple surface states could be used for achieving higher channel capacity. I can imagine how this could be true for Ref. [28] for 1D channels embedded in 2D, but a full 2D surface state embedded in 3D is not a "wire" that can carry information. Setting aside the point that this class of structures probably can't be realized for sub-cm scale wavelengths, I also don't see how using zone folding could give rise to higher channel capacity, even if there are nominally more surface states corresponding to higher Chern numbers.

Regarding the concept of higher capacity, the referee here raises a good point. In the following we intend to clarify this issue, better expressing our definition in terms on integrated quantities and channel capacity. On the one hand, the system with Chern number N supports N equifrequency loops on each surface. These N equifrequency loops are compressed into a folded BZ that is $1/N$ the size of the original BZ. In this sense, if we are interested in quantities integrated over the BZ, we haven't gained much with respect to the unit Chern number case, aligned by the reasoning of the referee. However, if we are interested in addressing states at a particular wavevector, which is a reasonable constraint in photonic systems, then the modulation has allowed us to address N chiral surface modes with momentum equivalent up to a reciprocal lattice vector. The situation is largely different in the unit Chern number case. In these terms, by higher channel capacity we mean the availability of multiple unidirectional surface channels, provided targeting a specific wavevector. Targeting specific momenta in photonic crystals is a possible task which can be achieved, for example, via plane-wave excitation by a directional source coupling to the surface modes of the passive photonic system [10, 11]. Moreover we also want to remark that the method developed based on Chern numbers by supercell modulation, setting aside the higher channel capacity aspects, also allows for the design of experimentally interesting photonic analogues of axionic responses [12, 13] that we are investigating as part of a future work.

Given the extreme difficulty of fabricating topologically non-trivial photonic crystals in 3D (especially with time-reversal symmetry breaking), my feeling is that a contribution more worthy of publication in Nature Communications would be one that constrains itself to the hard realities of current (or near-future) technology, and finds innovative solutions around these constraints.

* chiara.devescovi@dipc.org

† maiagvergniori@dipc.org

‡ aitzolgarcia@dipc.org

- [1] Zheng Wang, YD Chong, John D Joannopoulos, and Marin Soljačić. Reflection-free one-way edge modes in a gyromagnetic photonic crystal. *Physical review letters*, 100(1):013905, 2008.
- [2] Zheng Wang, Yidong Chong, John D Joannopoulos, and Marin Soljačić. Observation of unidirectional backscattering-immune topological electromagnetic states. *Nature*, 461(7265):772–775, 2009.
- [3] Kohei Kawabata, Ken Shiozaki, Masahito Ueda, and Masatoshi Sato. Symmetry and topology in non-hermitian physics. *Physical Review X*, 9(4):041015, 2019.
- [4] Jiewen Xiao and Binghai Yan. First-principles calculations for topological quantum materials. *Nature Reviews Physics*, 3(4):283–297, 2021.
- [5] NP Armitage, EJ Mele, and Ashvin Vishwanath. Weyl and dirac semimetals in three-dimensional solids. *Reviews of Modern Physics*, 90(1):015001, 2018.
- [6] Yihao Yang, Zhen Gao, Haoran Xue, Li Zhang, Mengjia He, Zhaoju Yang, Ranjan Singh, Yidong Chong, Baile Zhang, and Hongsheng Chen. Realization of a three-dimensional photonic topological insulator. *Nature*, 565(7741):622–626, 2019.
- [7] Long-Hua Wu and Xiao Hu. Scheme for achieving a topological photonic crystal by using dielectric material. *Physical review letters*, 114(22):223901, 2015.
- [8] Menglin LN Chen, Li Jun Jiang, Zhihao Lan, and EI Wei. Coexistence of pseudospin-and valley-hall-like edge states in a photonic crystal with c_3v symmetry. *Physical Review Research*, 2(4):043148, 2020.
- [9] Gui-Geng Liu, Zhen Gao, Peiheng Zhou, Qiang Wang, Yuan-Hang Hu, Maoren Wang, Chengqi Liu, Xiao Lin, Shengyuan A Yang, Yihao Yang, et al. Observation of weyl point pair annihilation in a gyromagnetic photonic crystal. *arXiv preprint arXiv:2106.02461*, 2021.
- [10] Sachin Vaidya, Jiho Noh, Alexander Cerjan, Christina Jörg, Georg Von Freymann, and Mikael C Rechtsman. Observation of a charge-2 photonic weyl point in the infrared. *Physical review letters*, 125(25):253902, 2020.
- [11] Minkyung Kim, Dasol Lee, Thi Hai-Yen Nguyen, Hee-Jo Lee, Gangil Byun, and Junsuk Rho. Total reflection-induced efficiency enhancement of the spin hall effect of light. *ACS Photonics*, 2021.
- [12] David Vanderbilt. *Berry Phases in Electronic Structure Theory: Electric Polarization, Orbital Magnetization and Topological Insulators*. Cambridge University Press, 2018.
- [13] Benjamin J Wieder, Kuan-Sen Lin, and Barry Bradlyn. Axionic band topology in inversion-symmetric weyl-charge-density waves. *Physical Review Research*, 2(4):042010, 2020.

REVIEWERS' COMMENTS

Reviewer #1 (Remarks to the Author):

In this revised manuscript, the authors made a serious attempt to address all my questions and concerns in a satisfactory manner. Thereby, I recommend it for publication in Nature Communications.

Reviewer #2 (Remarks to the Author):

The author's response and the revised manuscript have satisfactorily addressed all my previous concerns. In particular, the added explanation of the evolution of the surface states from Fermi arcs to Fermi loops is quite clear and insightful. Consequently, I would be happy to recommend it to be published in Nature Communications. My final minor suggestion is that the authors further correct their typos in the manuscripts, such as "is n is odd" → "if n is odd", "we the" → "we then", in line 5 and 6, respectively, in the right column of page 6.

Reviewer #3 (Remarks to the Author):

In response to my comments and questions, the authors made comprehensive explanations in their revised manuscript. I am sure that the present manuscript is ready for a publication in Nature communications.

Reviewer #4 (Remarks to the Author):

I appreciate the authors' thoughtful responses to my questions and comments. My original assessments still stand, but I emphasize that these had to do with experimental realization and device application, not the fundamental science aspect of the work. While I stand by my original recommendation not to publish due to these points, I can imagine that there may be interest from others in this work from that point of view.

Point-by-point response to the reviewers' 2nd comments for 'Cubic 3D Chern photonic insulators with orientable large Chern vectors'

Chiara Devescovi,^{1,*} Mikel García-Díez,^{2,1} Iñigo Robredo,^{3,1} María Blanco de Paz,¹ Jon Lasa-Alonso,^{4,5} Barry Bradlyn,⁶ Juan L. Mañes,⁷ Maia G. Vergniory,^{1,8,9,†} and Aitzol García-Etxarri^{1,8,‡}

¹*Donostia International Physics Center, 20018 Donostia-San Sebastián, Spain*

²*Physics Department, University of the Basque Country (UPV-EHU), Bilbao, Spain*

³*University of the Basque Country (UPV-EHU), Bilbao, Spain*

⁴*Centro de Física de Materiales, Paseo Manuel de Lardizabal 5, 20018 Donostia-San Sebastian, Spain*

⁵*Donostia International Physics Center, Paseo Manuel de Lardizabal 4, 20018 Donostia-San Sebastian, Spain*

⁶*Department of Physics and Institute for Condensed Matter Theory,*

University of Illinois at Urbana-Champaign, Urbana, IL, 61801-3080, USA

⁷*Physics Department, University of the Basque Country (UPV/EHU), Bilbao, Spain*

⁸*IKERBASQUE, Basque Foundation for Science, Maria Diaz de Haro 3, 48013 Bilbao, Spain*

⁹*Max Planck Institute for Chemical Physics of Solids, Dresden D-01187, Germany*

(Dated: October 25, 2021)

Dear Editors and Reviewers,

We thank the Editors and Referees for final the revision of the document. In the following, we respond point by point to all of their comments. Our point-by-point responses are highlighted in blue and are provided together with the original comments reproduced verbatim.

REVIEWER #1

In this revised manuscript, the authors made a serious attempt to address all my questions and concerns in a satisfactory manner. Thereby, I recommend it for publication in Nature Communications.

We thank the reviewer for the questions which, we believe, helped us improving the quality of our manuscript substantially.

REVIEWER 2

Reviewer 2 (Remarks to the Author): The author's response and the revised manuscript have satisfactorily addressed all my previous concerns. In particular, the added explanation of the evolution of the surface states from Fermi arcs to Fermi loops is quite clear and insightful. Consequently, I would be happy to recommend it to be published in Nature Communications. My final minor suggestion is that the authors further correct their typos in the manuscripts, such as "is n is odd" -> "if n is odd", "we the" -> "we then", in line 5 and 6, respectively, in the right column of page 6.

We thank the reviewer for his/her comments and response. About the highlighted typos, we have corrected them in the final manuscript.

REVIEWER 3

In response to my comments and questions, the authors made comprehensive explanations in their revised manuscript. I am sure that the present manuscript is ready for a publication in Nature communications.

We thank the reviewer for their reply and for acceptance of the explanations provided.

REVIEWER 4

I appreciate the authors' thoughtful responses to my questions and comments. My original assessments still stand, but I emphasize that these had to do with experimental realization and device application, not the fundamental science aspect of the work. While I stand by my original recommendation not to publish due to these points, I can imagine that there may be interest from others in this work from that point of view.

We thank the reviewer for the time dedicated to reviewing the manuscript and for the appreciation about our explanations.

* chiara.devescovi@dipc.org

† maiagvergniori@dipc.org

‡ aitzolgarcia@dipc.org